# High-resolution bathymetry models for the Lena Delta and Kolyma Gulf coastal zones

Matthias Fuchs[1], Juri Palmtag[2], Bennet Juhls[1,3], Paul Overduin[1], Guido Grosse[1,4], Ahmed Abdelwahab[1,4], Michael Bedington[5], Tina Sanders[6], Olga Ogneva[1], Irina V. Fedorova[7], Nikita S. Zimov[8], Paul J. Mann[2], Jens Strauss[1]

[1]Alfred Wegener Institute Helmholtz Centre for Polar and Marine Research, Potsdam, Germany
[2]Department of Geography & Environmental Sciences, Northumbria University, Newcastle upon Tyne, UK
[3]Department of Earth Sciences, Institute for Space Sciences, Freie Universität Berlin, Berlin, Germany
[4]Institute of Geosciences, University of Potsdam, Potsdam, Germany
[5]Plymouth Marine Laboratory, Plymouth, UK
[6]Helmholtz-Zentrum Hereon, Institute for Carbon Cycles, Geesthacht, Germany
[7]St. Petersburg State University, Institute of Earth Science, St. Petersburg, Russia
[8]North-East Scientific Station, Pacific Institute for Geography, Far-East Branch, Russian Academy of Sciences, Cherskiy, Russia

*Correspondence to*: Matthias Fuchs (matthias.fuchs@awi.de)

**Abstract.** Arctic river deltas and deltaic near-shore zones represent important land-ocean transition zones influencing sediment dynamics and nutrient fluxes from permafrost-affected terrestrial ecosystems into the coastal Arctic Ocean. To accurately model fluvial carbon and freshwater export from rapidly changing river catchments, as well assessing impacts of future change on the Arctic shelf and coastal ecosystems, we need to understand the sea floor characteristics and topographic variety of the coastal zones. To date, digital bathymetrical data from the poorly accessible, shallow and large areas of the eastern Siberian Arctic shelves are sparse. We have digitized bathymetrical information for nearly 75,000 locations from large-scale (1:25,000 – 1:500,000) current and historical nautical maps of the Lena Delta and the Kolyma Gulf region in Northeast Siberia. We present the first detailed and seamless digital models of coastal zone bathymetry for both delta/gulf regions in 50 m and 200 m spatial resolution. We validated the resulting bathymetry layers using a combination of our own water depth measurements and a collection of available depth measurements, which showed a strong correlation (r > 0.9). Our bathymetrical models will serve as an input for a high-resolution coupled hydrodynamic-ecosystem model to better quantify fluvial and coastal carbon fluxes to the Arctic Ocean but may be useful for a range of other studies related to Arctic delta and near-shore dynamics such as modelling of submarine permafrost, near-shore sea ice, or shelf sediment transport. The new digital high-resolution bathymetry products are available on the PANGAEA data set repository (Fuchs et al. 2021a, b). Likewise, the depth validation data is available on PANGAEA as well (Fuchs et al., 2021c).

## 1 Introduction

The bathymetry of the Arctic Ocean at the mouths of major rivers is at the locus of interactions between land-to-ocean sediment fluxes, fluvial discharge, alongshore currents, and sea ice dynamics. It exerts control on these processes and is also shaped by them. Therefore, it is imperative to establish a baseline by which to compare future bathymetrical changes especially since processes at the land-ocean interface are changing rapidly due to climate-change. For example, Arctic river discharge is increasing (Peterson et al., 2002; McClelland et al., 2006; Haine et al. 2015; Holmes et al., 2015; Brown et al., 2019), resulting in altered sediment, nutrient and organic carbon loads exported from rapidly changing river catchments into near-shore regions (e.g. Rachold et al., 2000; Gordeev, 2006; Tank et al., 2016; Wild et al., 2019) with unclear effects on the Arctic shelf and ocean ecosystems (Mann et al, 2022; Sanders et al. 2022; Polimene et al, in review). In particular, Arctic deltas will be affected by climate change-induced increase of permafrost temperatures (Biskaborn et al. 2019), changing sea ice distribution (Stroeve and Notz, 2018), sea level rise (Box et al., 2018), increasing storm surges (Vermaire et al. 2018), coastal erosion rates (Jones et al., 2020), and warming water temperatures in the Arctic coastal systems (Timmermans and Labe, 2020). Therefore, better baseline data are needed for Arctic deltas and their often very shallow subaquatic near-shore zones to quantify and model the effects of climate change induced disturbances upon these sensitive environments.

Regional scale models have been shown to be sensitive to changes in bathymetry, particularly in near-shore zone and shallow areas, as these significantly affect both the distribution of tidal currents and stratification dynamics (Fofonova et al., 2013; 2014; Shulman et al., 2013; Anand & Kumar, 2018; Rasquin et al., 2020). Furthermore, low-resolution bathymetry products can entirely obscure small-scale near-shore features and processes, such as vestigial river channels, which may nevertheless represent significant local oceanographic features (Lee and Valle-Levinson, 2012; Janout et al. 2017; Ye et al., 2018) or coastal ground water dynamics, which might become even more important in the future as permafrost thaws (Connolly et al., 2020).

However, deltaic and coastal zone morphologies of Arctic rivers have been rarely studied so far, primarily due to difficulties in accessing these extensive yet often shallow and highly changeable regions - resulting in a general lack of available data. While coarse resolution (~200 m) bathymetry products are available for Arctic shelves, e.g. the International Bathymetrical Chart of the Arctic Ocean (IBCAO, Jakobsson et al., 2020), detailed digital bathymetry data are missing for Arctic delta and coastal areas. Bathymetrical data of these environments are important to study land-to-ocean processes and thus a much-needed input e.g. for models estimating the river outflow through deltas and the pathways of organic matter transport, deposition and transformation in the coastal zones of the Arctic Ocean. A range of other modelling subjects also require high resolution coastal zone bathymetrical data such as submarine permafrost, nearshore sea ice dynamics, coastal erosion, storm or tidal surges, or shelf sediment transport and deposition patterns.

Therefore, the aim of this study was to create two detailed, high-resolution digital coastal zone bathymetry data sets derived from analogue nautical maps for the Lena Delta and Kolyma Gulf regions in northeast Siberia. The data set will serve as a model input for a planned high-resolution implementation of the coupled FVCOM-Arctic European Regional Shelf Seas

Ecosystem (Arctic-ERSEM) Model (Butenschön et al. 2016; Bedington et al. 2021). Besides providing an important model input, a high-resolution bathymetry for coastal zones offers insights into the continuation of river channels into the non-terrestrial part of the delta/gulf as well as erosion and accumulation zones in the land-ocean interface.

## 2 Material and Methods

For this data set, we used 28 analogue nautical charts for creating a digital high-resolution bathymetry data set for two coastal areas of river mouths in the Laptev and East Siberian Sea (Figure 1). Our first data set included the coastal zone bathymetry of the Lena River Delta and the southern Laptev Sea shelf (up to 250 km north of the Lena Delta). The second data set presents the river and estuary coastal zone bathymetry from the Kolyma River Gulf region (up to 70 km offshore). In addition, we use water depth information from our own field-measured conductivity temperature and depth (CTD) observations as well as compiled water depth measurements synthesized from the PANGAEA (https://pangaea.de) data archive for both regions as a cross-validation data set for our digital bathymetry.

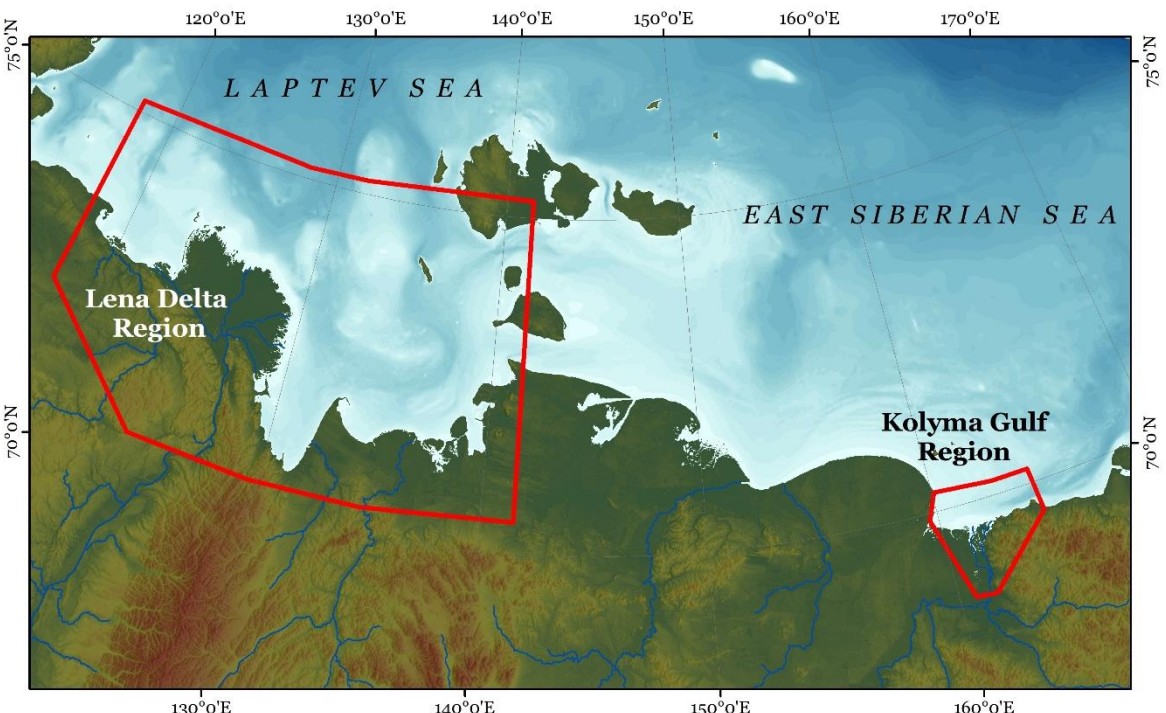

**Figure 1:** Study areas (red frames) of the Lena Delta and Kolyma Gulf regions. Background image is the IBCAO v4, 200 m spatial resolution raster map (Jakobsson et al., 2020).

## 2.1 Nautical Maps

As basis for the creation of a digital bathymetrical raster, 28 Russian nautical charts in the scales of 1:25,000 – 1:500,000
(Table 1+2, Appendix figure A1+A2) were digitized. The maps were acquired from EastView Geospatial as georeferenced raster images. The projection of the maps was Gauss-Krueger (Datum Pulkovo 1942). According to the map legends, the original soundings and topographic surveys for these maps were carried out since the 1940s (Table 1 and 2) by the Department of Navigation and Oceanography, Russian Federation Ministry of Defense. Water depth points are given in meters and are in most cases reduced to mean sea level. Digitization of bathymetric information in the maps was manually conducted in ArcGIS
TM version 10.6 and included individual water depths (points) as well as isobaths (lines), which then served as input data for our bathymetrical model. Nautical charts served previously as primary data for bathymetrical data sets in regions where input data are scarce, such as for the IBCAO (Jakobsson et al., 2020) or the International Bathymetrical Chart of the Southern Ocean (IBCSO) (Arndt et al., 2013). For shallow shelf zones (< 200 m), nautical charts have been shown to be a valid data sources for more accurate bathymetrical products for coastal zones in the Indian Ocean (Sindhu et al., 2007).

For the Lena Delta a total of 50,828 water depth points and 720 isobath lines were digitized from 15 nautical maps (Table 1). For the Kolyma Gulf, a total of 24,126 water depth points and 1,053 isobath lines were digitized from 13 nautical maps (Table 2). The different nautical charts and various scales within a region partially overlapped with each other leading to a denser, non-uniform point cloud of depth measurements (for the extent of the nautical charts, see Appendix A, figure A1+A2). In most of the cases, depth measurements of overlapping parts were matching. If points were not congruent (e.g. due
to different survey times or different map compilation dates, see Table 1 + 2), the location of the depth measurement from the higher resolution map was chosen. Digitization of the isobaths followed a similar strategy. Offsets in isobaths lines from different charts were corrected by choosing the isobaths of the higher resolution map.

 **Table 1:** Map charts from the Lena Delta region

| Map Chart ID | Map Title (region) | Map Scale | Map Edition | Years of soundings and topographic surveys | Comment |
|---|---|---|---|---|---|
| 11142 | Approaches to the Deltas of the Lena and Olenenk rivers | 1:500`000 (at 71.30°N) | 1994 | 1945-1946, 1961-1965, 1967-1970, 1972-1980; 1982-1988 | Depths on the chart westward of the meridian 119°30' are reduced to astronomical tide and those eastward of it are reduced to mean sea level |
| 11141 | From the Lena Delta to Sannikov and Dmitriy Laptev strait | 1:500`000 (at 71.30°N) | 1994 | 1952, 1955-1961, 1963-1972, 1974, 1975, 1981, 1985-1988 | Depths are reduced to mean sea level |
| 11140 | From Tiksi harbour to the Dmitriy Laptev Strait | 1:500`000 (at 71.30°N) | 1995 | 1938, 1939, 1943-1962, 1964-1972, 1995 | Depths are reduced to mean sea level |
| 13416 | Eastern side of Olenekskyi Bay | 1:100`000 (at 75°N) | 1999 | 1945, 1968-1970, 1980, 1982, 1983 | Depths are reduced to mean sea level |
| 13411 | Northwestern region of the Lena River Delta | 1:100`000 (at 75°N) | 1996 | 1945, 1946, 1953, 1954, 1964, 1967, 1968 | Depths are reduced to mean sea level |
| 13418 | From the Dunay Islands to Cape Doktorskiy | 1:100`000 (at 75°N) | 1998 | 1961, 1963, 1966, 1967, 1972, 1973 | Depths are reduced to mean sea level |
| 13419 | From Cape Doktorskiy to the Khastyr-Tördün-Bölköydörö Islands | 1:100`000 (at 75°N) | 2000 | 1953, 1954, 1961, 1963, 1973-1975 | Depths are reduced to mean sea level |
| 13420 | From the Khastyr-Tördün-Bölköydörö Islands to the Bolshaya Trofimovskaya channel | 1:100`000 (at 75°N) | 1995 | 1953, 1961, 1963, 1967, 1974, 1975 | Depths are reduced to mean sea level |
| 13421 | From the Bolshaya Trofimovskaya channel to the Ispolatov channel | 1:100`000 (at 75°N) | 1995 | 1953, 1954, 1956, 1961, 1962, 1965, 1967, 1972, 1975-1977 | Depths are reduced to mean sea level |
| 13422 | Approaches to Tiksi Bay and the Bykovskaya channel of the Lena river | 1:100`000 (at 75°N) | 2014 | 1940, 1942, 1943-1945, 1954-1959,1961, 1963, 1964, 1965, 1967, 1968, 1970, 1972, 1977, 1978, 1982, 1984, 1988, 1994, 1999 | Depths are reduced to mean sea level |
| 13423 | Southern part of the Buor-Khaya Bay | 1:100`000 (at 75°N) | 2001 | 1944, 1945, 1949, 1963 | Depths are reduced to mean sea level |
| 13424 | Northeastern part of the Buor-Khaya Bay | 1:100`000 (at 75°N) | 2006 | 1944, 1945, 1954, 1962, 1963, 1965, 1967, 1970, 1972 | Depths are reduced to mean sea level |
| 15465 | The entrance to the Bykovskaya channel | 1:50`000 (at 75°N) | 2014 | 1941, 1954-1956, 1977, 1982, 1983, 1999 | Depths are reduced to mean sea level |
| 15466 | Bykovsky fairway and Nelov Bay | 1:50`000 (at 75°N) | 1999 | 1941, 1943, 1954-1956, 1961, 1962, 1970, 1971, 1977, 1982, 1988, 1994 | Depths are reduced to mean sea level |
| 15467 | Routes to the harbour of Tiksi | 1:50`000 (at 75°N) | 1995 | 1941-1965 | Depths are reduced to mean sea level |

**Table 2**: Map charts from the Kolyma Gulf region

| Map Chart ID | Map Title (region) | Map Scale | Map Edition | Years of soundings and topographic surveys | Comment |
|---|---|---|---|---|---|
| 14411 | From Cape Krestovskiy to Protoka Strait | 1:100`000 (at 69°N) | 1995 | 1956, 1975, 1966, 1976, 1978, 1980, 1982, 1986-1989 | Depths are reduced to mean sea level |
| 14412 | Approaches to the Kolyma River Delta | 1:100`000 (at 69°N) | 1999 | 1956, 1958, 1965-1967, 1969, 1973, 1974, 1976-1980, 1982-1989 | Depths are reduced to mean sea level |
| 14413 | Bukhta Ambarchik to Mys Bol´shoy Baranov | 1:100`000 (at 69°N) | 1999 | 1956, 1958, 1960, 1976-1978, 1982-1987, 1989 | Depths are reduced to mean sea level |
| 16408 | Approaches to Pokhodsk | 1:25`000 | 1994 | 1966, 1969, 1982 | Depths north of 69°08`N are reduced to mean sea level, depths south of 69°08`N are reduced to lower mean navigational level |
| 16409 | Central part of Pokhodskaya Protoka | 1:25`000 | 1995 | 1964-1966, 1969, 1982 | Depths are reduced to mean sea level |
| 16410 | North part of Pokhodskaya Protoka | 1:25`000 | 1994 | 1964-1966, 1969, 1982, 1986 | Depths are reduced to mean sea level |
| 16411 | Mouth of Pokhodskaya Protoka | 1:25`000 | 1995 | 1956, 1965, 1966, 1969 1979, 1980, 1986 | Depths are reduced to mean sea level |
| 16412 | Mys Medvezhiy to Ostrov Gusmp | 1:50`000 (at 69°N) | 1994 | 1965-1967, 1973, 1974, 1976-1980, 1982, 1983, 1984-1987, 1989 | Depths are reduced to mean sea level |
| 16413 | Ostrov Gusmp to Mys Kolymskaya Strelka | 1:50`000 (at 69°N) | 1996 | 1964-1966, 1969-1974, 1976, 1977, 1981-1989, 1990 | Depths north of 69°08`N are reduced to mean sea level, depths south of 69°08`N are reduced to lower mean navigational level which is lower by 1 m. |
| 19423 | Mys Medvezhiy to Mys Obryvistyy | 1:25`000 | 1999 | 1956, 1964, 1965, 1976-1979, 1982-1987, 1989 | Depths are reduced to mean sea level |
| 19424 | Mys Obryvistyy to Kur`ishka | 1:25`000 | 1996 | 1956, 1964, 1965, 1973, 1974, 1983, 1984-1987, 1989 | Depths are reduced to mean sea level |
| 19425 | Kur`ishka to Mys Verkhnekabachkovskiy | 1:25`000 | 2008 | 1964, 1965, 1971-1973, 1977, 1983, 1984, 1986, 1987, 1989 | Depths are reduced to mean sea level |
| 19426 | Mys Verkhnekabachkovskiy to Mis Filipposvskaya Strelka | 1:25`000 | 2008 | 1965, 1969-1971, 1976, 1977, 1982-1988 | Depths north of 69°08`N are reduced to mean sea level, depths south of 69°08`N are reduced to lower mean navigational level which is lower by 1 m. |

**2.2 Creation of the bathymetrical models based on the Topo to Raster interpolation method**

For the interpolation of the digitized water depth points in combination with the isobath lines, we used the Esri ArcGIS 10.6 Topo to Raster (TTR) tool. TTR is a spatial interpolation method based on the ANUDEM (Australian National University Digital Elevation Model) from Hutchinson (1989) with the aim to create an accurate and hydrologically correct elevation model. We determined the outer boundary seawards by the extent of the nautical maps. Therefore, no extrapolation towards the outer East Siberian Sea and Laptev Sea outside the water depth points is calculated. For the Kolyma Gulf region, the Global Surface Water Layer (Pekel et al., 2016) was used to delineate the water area. However, for the Lena Delta region this proved not to be feasible due to the many small banks and islands. Therefore, the water area was derived from a water index based on twelve Landsat 8 scenes. More information about the water area determination is compiled in the appendix B. The following input data were used in the TTR tool for the models:

- Digitized point layer from nautical maps as "point elevation"
- Digitized isobath lines from nautical maps as "contour"
- The water area sets the boundary of the bathymetrical model.

In addition, we set the primary data input type to points and set the parameter for drainage enforcing to "no-enforce" in order to allow sinks within the bathymetrical model, since hydrological enforcing would only apply to runoff systems in terrestrial elevation models but not to bathymetrical models. We ran the tool with 20 iterations and set the maximum height to 0.0 m so that the tool did not extrapolate above mean sea level and the resulting product only contains depths below sea level. Model runs were executed with a pixel size of 50 m and 200 m resulting in two products, a $TTR_{50}$ bathymetrical data set and $TTR_{200}$ bathymetrical data set for both regions.

The advantage of the TTR tool is that it allows usage of points (e.g. the digitized depth measurements) as well as line features (e.g. the digitized isobaths lines) as input data to produce the bathymetrical model. This was particularly useful in the preserving channel outlets at the transition from river to the near-shore zone in order to avoid artificial sinks caused by the interpolation from points only. Without isobaths integration, the TTR models created beaded, artificial sinks in nearly all of the Kolyma and Lena river channels inside the delta/gulf as well as on the river-coastal transition zones.

**2.3 Model validation and comparison to existing bathymetry products**

**2.3.1 Field measurements from the near-shore**

Combined conductivity, temperature, and depth (CTD) data were collected on three field trips as part of the 'Changing Arctic Carbon cycle in the cOastal Ocean Near-shore (CACOON)' project during spring (Strauss et al., 2021) and summer 2019 (Fuchs et al., 2021d) from the Lena Delta and Kolyma Gulf region (Figure 2). We obtained CTD measurements in the Lena Delta using a handheld Sontek™ CastAway CTD device (for measurement accuracies see appendix C) with an integrated GPS. With this compact device, we were able to obtain data from small dinghy boats also in shallow areas where larger vessels have

no access. At each location, the CTD was lowered using additional ballast to ensure the CTD reached the sea (or river) floor. CTD measurements in the Kolyma Gulf region were made with a HYDROLAB HL7 multiparameter probe (OTT Hydromet).

For our CTD measurements, we targeted specifically the coastal near-shore and the transition from river waters to open sea, since these regions have only sparse coverage in other data sets (see chapter 2.3.2), often caused by the difficulty to access these shallow waters. With our unique and valuable data set, we complement other CTD data set from these regions (e.g. Hölemann et al. 2020) and fill a critical gap in the coastal near-shore zone by specifically targeting the mouth of Arctic river channels. Our CTD metadata and data are available on the PANGAEA database (Fuchs et al. 2021c) from the Lena Delta

near-shore zone and at the British Ocean Data Center (BODC) from the Kolyma main channel and the Kolyma near-shore zone (Palmtag and Mann, 2021; Palmtag et al., 2021).

In the Lena Delta region, CTD measurements were collected on 31 locations in the Sardakhskaya channel spanning from Stolb Island at the apex of the Lena Delta to 80 km offshore in the Laptev Sea (Figure 2a). Depth measurements ranged from 1.9 m (CAC19-C) to 21.5 m (CAC19-S-09) in the Lena Delta region. Aside from depth measurements, pressure (dbar),

water temperature (°C), conductivity (mS/cm), conductance (mS/cm), salinity (practical salinity scale), sound velocity (m/s), and density ($kg/m^3$) were also recorded and are provided with the data set.

In the Kolyma Gulf region, CTD measurements were collected during seven different trips spanning from the freshet (11 June 2019) to late summer (2 September 2019). In total, 67 profiles were measured along a 140 km long transect starting from Cherskiy at the apex of the Kolyma Gulf northwards to the East Siberian Sea (Figure 2b + 2c). The HYDROLAB HL7

multiparameter probe used for these measurements recorded data from a wide range of parameters (specific conductivity (mS/cm), turbidity (NTU), barometric pressure (mmHg), dissolved oxygen (mg/L), depth (m), water temperature (°C), density ($kg/m^3$), salinity (psu) and chlorophyll-*a* concentration (µg/L)).

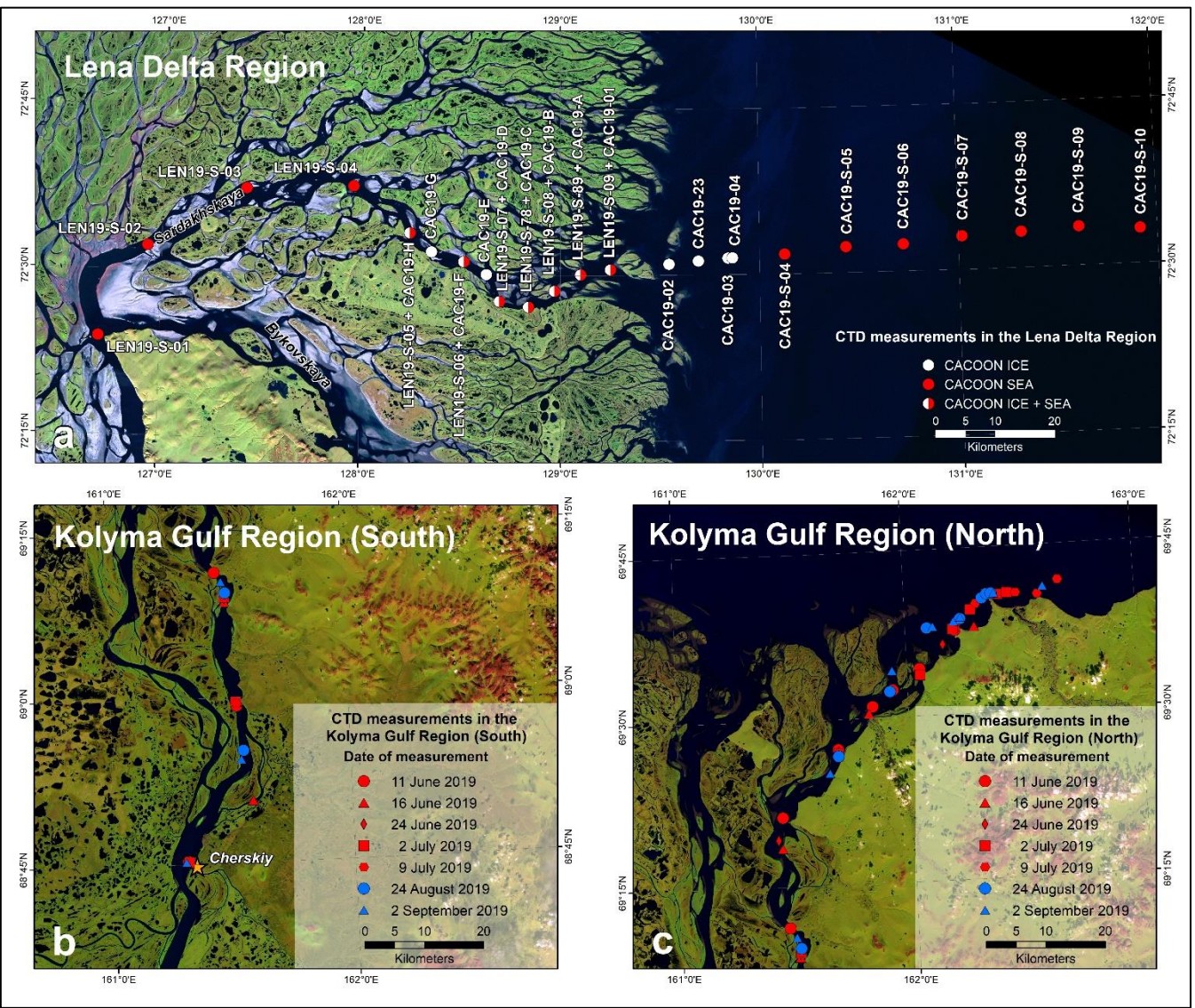

**Figure 2:** a) CTD measurement locations in the Lena Delta region during the CACOON 2019 expeditions (Fuchs et al., 2021c.; Fuchs et al., 2021d; Strauss et al., 2021, appendix Table C1) starting from Stolb Island (LEN19-S-01) passing through the Sardakhskaya main channel to 80 km offshore at CAC19-S-10 (background image: Landsat-5 mosaic (band combination 5, 4, 3) including scenes from 2009 and 2010; Landsat-5 image courtesy of the U.S. Geological Survey). CTD data on the CACOON Ice expeditions were collected from 30 March to 4 April 2019 and CTD data on the CACOON Sea expedition were collected from 3 August to 9 August 2019. b) CTD measurement locations in the Kolyma Gulf region in the southern (b) and in the northern part (c) (Palmtag and Mann, 2021) (background image: Landsat 8 mosaic (band combination 6, 5, 3) including scenes from 2019; Landsat-8 image courtesy of the U.S. Geological Survey). Orange star shows the city of Cherskiy. Campaigns in both regions aimed to cover the coastal near-shore and the transition zone from riverine to marine systems.

### 2.3.2 Additional archived data for model validation

Complementary to our own collected CTD data, we synthesized 660 historical and publicly available depth measurements
from 14 additional data sets, available in the PANGAEA archive for the Lena Delta region for an additional validation of our
TTR bathymetry model. From 1994 – 2014, depth measurements were acquired by the Transdrift campaigns
(https://www.transdrift.info/de). Water depths from the Transdrift campaigns I to IX, XII, XVII, XIX, XXI, and XXII (Bauch
et al., 2018; Bauch et al., 2009; Janout et al., 2019a, 2019b; Hölenmann et al., 2020; Transdrift Community Members, 2009a,
2009b, 2009c, 2009d, 2009e, 2009f, 2009g, 2009h, 2009i) were measured with a Seabird SBE 19+ CTD profiler and we
compared these to the $TTR_{50}$ Lena Delta bathymetrical model. In addition to the Transdrift data, further CTD data available
on PANGAEA were used (e.g. Bussmann, 2013; Dubinenkov et al., 2015; Gonçalves-Araujo et al., 2015; Wagner et al., 2012;
Wetterich et al., 2011). Combined with our own CTD measurements (see chapter 2.3.1), this resulted in 671 points for
validation of the Lena Delta region bathymetrical model (Figure 3). For the Kolyma Gulf region, the available depth
measurements for comparison are sparse. Therefore, we compared our bathymetrical model only to our own collected CTD
data points. In order to further validate the quality of the Kolyma Gulf region $TTR_{50}$ model, we executed a cross-validation
where we removed 1,030 random points from the model run for validation (4.3% of the available points). The model was then
run with the remaining ~23,000 points and the output was compared to the 1,030 omitted points.

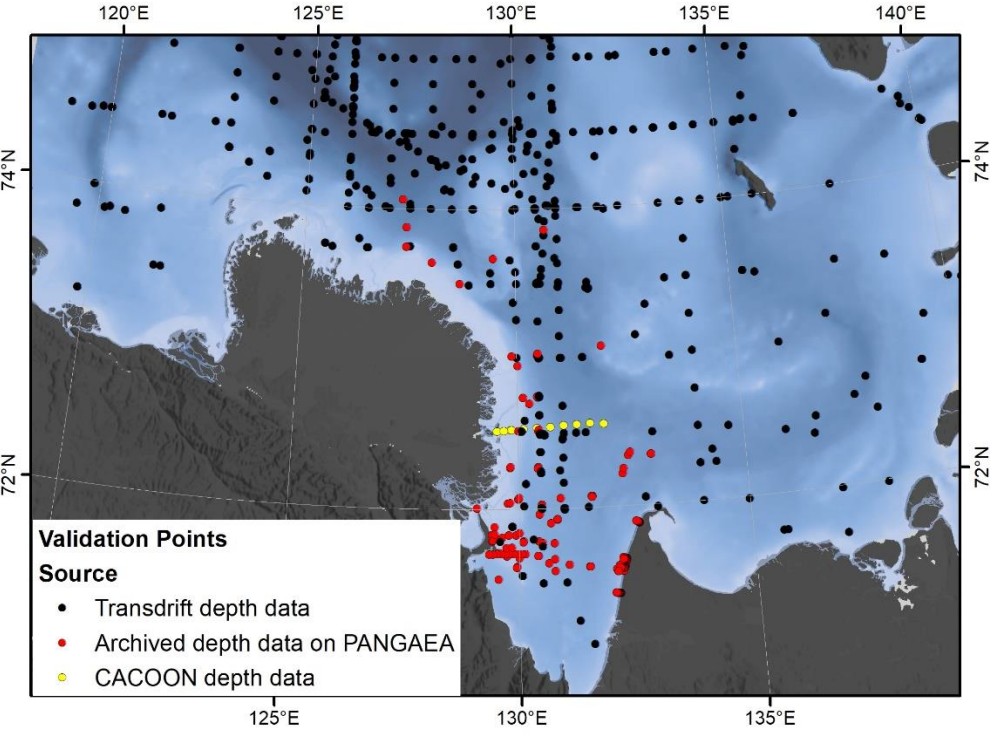

      **Figure 3:** Location of validation points for the Lena Delta region bathymetrical model

## 2.3.3 Comparison of the bathymetrical models to the IBCAO

In addition to the validation of our $TTR_{50}$ bathymetry models with CTD data, we compared the $TTR_{200}$ models to the International Bathymetric Chart of the Arctic Ocean (IBCAO), which provides a bathymetrical model for the Arctic above 64°N latitude (Jakobsson et al., 2020). The latest version (v4) of the IBCAO dataset was released in July 2020 with a 200 m grid size at its highest spatial resolution. The compilation of the IBCAO v4 includes different base data, such as single-beam and multi-beam echo-soundings or digitized contour/isobaths lines from nautical charts (Jakobsson et al., 2020). While there are numerous bathymetrical data available from the central Arctic Ocean from numerous international ship campaigns, such data are much more sparse for the often very shallow Arctic shelves and in particular the coastal zone in the Russian Arctic, resulting in high uncertainties for the IBCAO v4 accuracy for these shelf areas. In order to assess how well our digital bathymetrical products compare with the existing IBCAO v4 dataset for the Lena and Kolyma coastal zones, we calculated vertical difference models. We used our 200 m resolution TTR bathymetrical model for both the Lena Delta and Kolyma Gulf region and subtracted the IBCAO v4 200 m raster from the $TTR_{200}$ models to identify zones of large differences between these two data products.

## 3 Bathymetrical models of the Lena and Kolyma near-shore zone

For both study areas, we executed TTR model runs for a 50 m and 200 m resolution bathymetrical model. Both final data sets are available on the PANGAEA data set repository as geotiff raster files (Fuchs et al. 2021a; 2021b). In addition, the published data sets include the depth point input data, the isobaths line input data and the water area polygon input data for both, the Lena Delta and the Kolyma Gulf region, in shapefile format. The data sets can be accessed with the following links: https://doi.pangaea.de/10.1594/PANGAEA.934045 (Fuchs et al., 2021a) and https://doi.pangaea.de/10.1594/PANGAEA.934049 (Fuchs et al., 2021b)

## 3.1 Lena Delta region bathymetry model

The Lena Delta region bathymetry model (Figure 4) includes more than 50,000 depth points derived from nautical charts and covers an area of 232,700 km$^2$ stretching from Cape Mamontov Klyk in the western Laptev Sea to Kotelny Island in the New Siberian Islands. The highest point density with an average of nearly five depth measurements per square kilometre was found in the Tiksi Bay area and around Bykovsky Peninsula (see appendix Figure D1). Overall, water depths in the bathymetrical model region ranged from 0 to 55.0 m with an average depth of 17.6 m ± 9.0 m. Our TTR bathymetry models particularly show the transition and continuation of the Lena Delta main channels into the Laptev Sea (Figure 4b + 4c), providing important indications of the direction and volume of water outflow from the Lena River into the Laptev Sea.

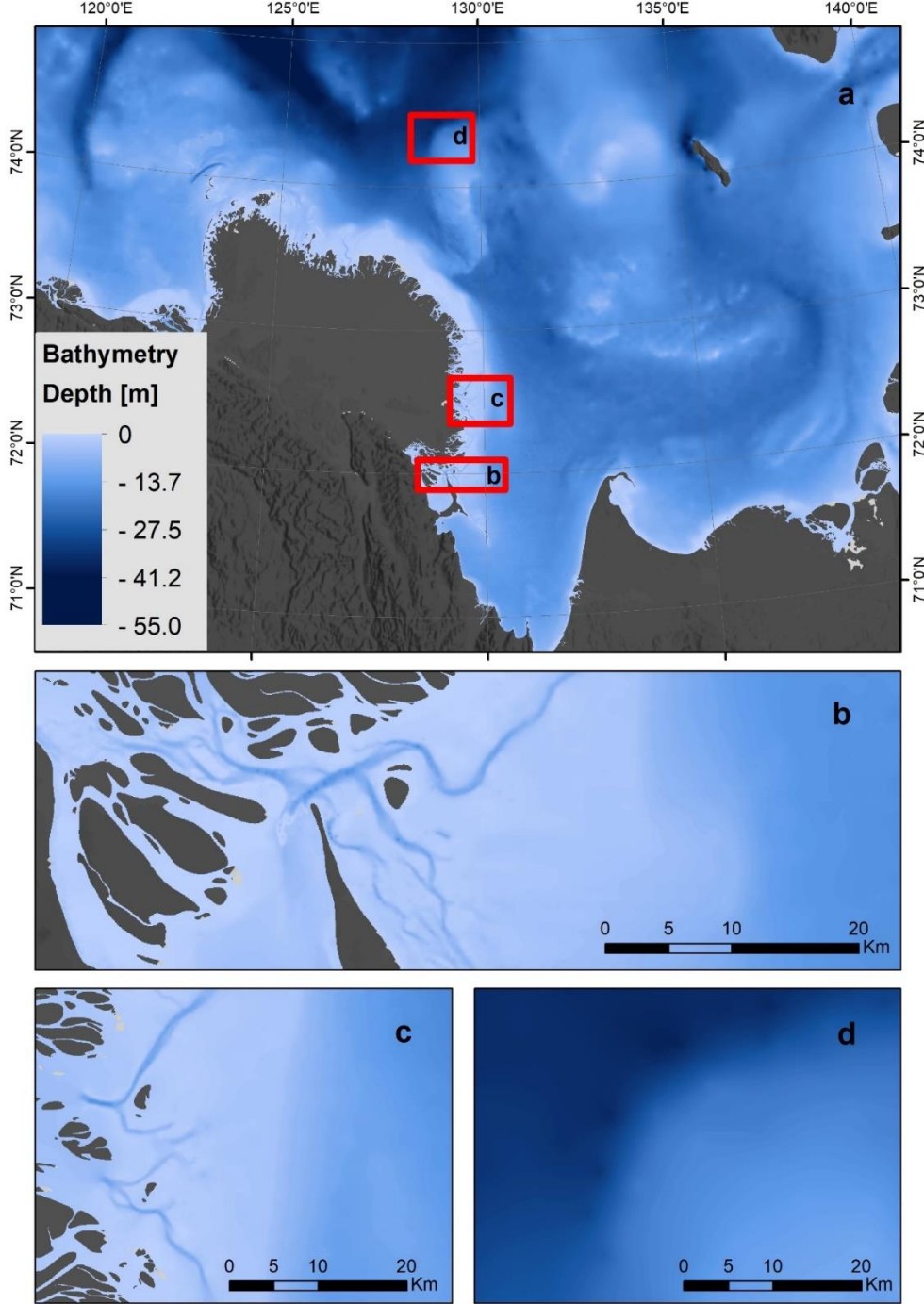

**Figure 4:** a) Topo to raster modelled bathymetry (TTR$_{50}$) of the Lena Delta region with detailed map inlets showing the underwater channels at the outlet of the Bykovskaya channel (b), which became visible due to the inclusion of the isobaths

lines. c) shows the near-shore area of the Sardakhskaya channel and map inlet d) shows the transition from a shallow area into a deep open water area 80 km north of the Lena Delta.

## 3.2 Kolyma Gulf region bathymetry model

The Kolyma Gulf region bathymetry model (Figure 5a) includes more than 24,000 depth points derived from nautical charts and covers an area of 12,100 km$^2$ starting from about 5 km downstream of the city of Cherskiy to 70 km into the East Siberian Sea. Water depths in the bathymetrical model ranged from 0 to 32.5 m with an average depth of 10.4 m $\pm$ 6.6 m. The bathymetry shows the continuation of the Kolyma main channels into the near-shore (Figure 5b) and the transition to deeper coastal areas. The highest point density with an average of 17 measurements per square kilometre was found in the Kolyma river main channel (see appendix Figure D2). The high point resolution in the channels also shows a detailed channel morphology. Two deep river channels are present inside the Kolyma Gulf, passing through both of the main channels. Although they have varying depth (Figure 5c + d) they indicate paths of predominant water flow, also during wintertime when the Kolyma river is frozen.

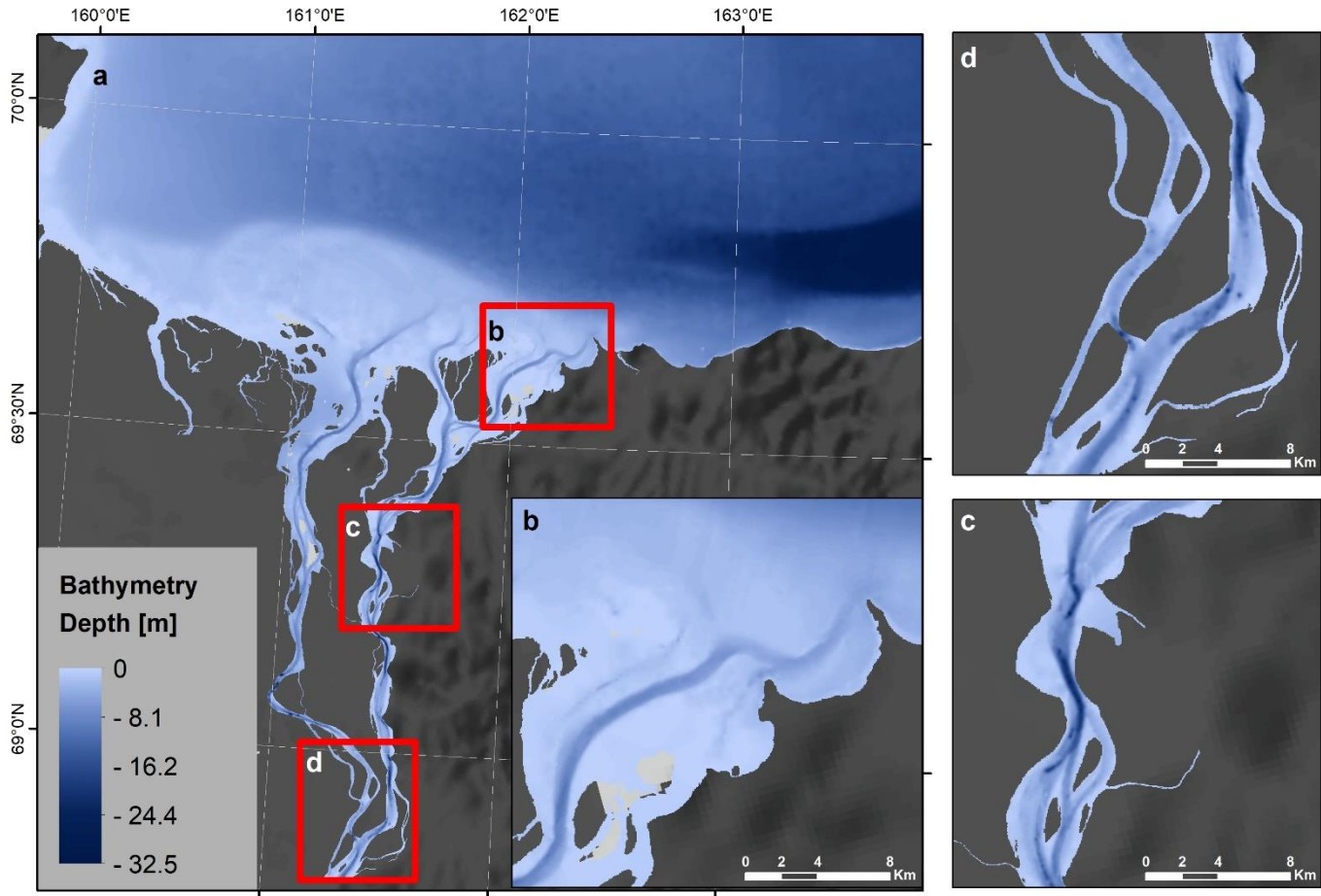

**Figure 5:** a) Topo to raster modelled bathymetry (TTR$_{50}$) of the Kolyma Gulf region with three zoomed-in subsets on the Kolyma river mouth (b) in the central Kolyma Gulf showing one of the deepest parts of the main river channel (c) and at the apex point of Kolyma Gulf region (d), where the two main channels split.

## 4 Validation, comparison, and limitations

The Lena Delta and Kolyma Gulf region bathymetry were validated using our own depth field data (see section 2.3.1) and
235 synthesized depth data (including Transdrift CTD data; see section 2.3.2), which were not included in the TTR$_{50}$ model calculations. In addition, we compared our TTR$_{200}$ models with the IBCAO v4 200 m (Jakobsson et al., 2020) bathymetry to detect differences between these two data sets.

### 4.1 Validation of the Lena Delta region bathymetry based on CTD data

The validation of the Lena Delta region bathymetry (TTR$_{50}$) with the 671 CTD points for validation (Figure 3) showed a good
agreement (Figure 6). The Pearson`s correlation coefficient between all the CTD points and the Lena bathymetry is 0.98 ($p <$

0.001) and while the Transdrift data mostly cover the deeper parts of the bathymetrical model, the depth data from smaller campaigns (e.g. Bussmann, 2013; Dubinenkov et al., 2015; Fuchs et al., 2021c; Strauss et al., 2021) cover the shallower parts (< 20 m depth) of the Lena bathymetry. The aim of the $TTR_{50}$ bathymetry model was specifically to target the shallow near-shore area, therefore our own collected depth data in this area help to validate the models. In addition, the synthesized data sets included the largest amount of validation points in water depths less than 30 m (Figure 6a), which show a good agreement with the modelled data for these locations (Figure 6b). A few validation points show a larger deviation from the model (> 5m). These points may indicate real bathymetric features such as small scale variabilities in the sea floor, which are not captured by $TTR_{50}$ bathymetry. The location of these points including the deviation from the $TTR_{50}$ bathymetry are presented in the appendix E (figure E1).

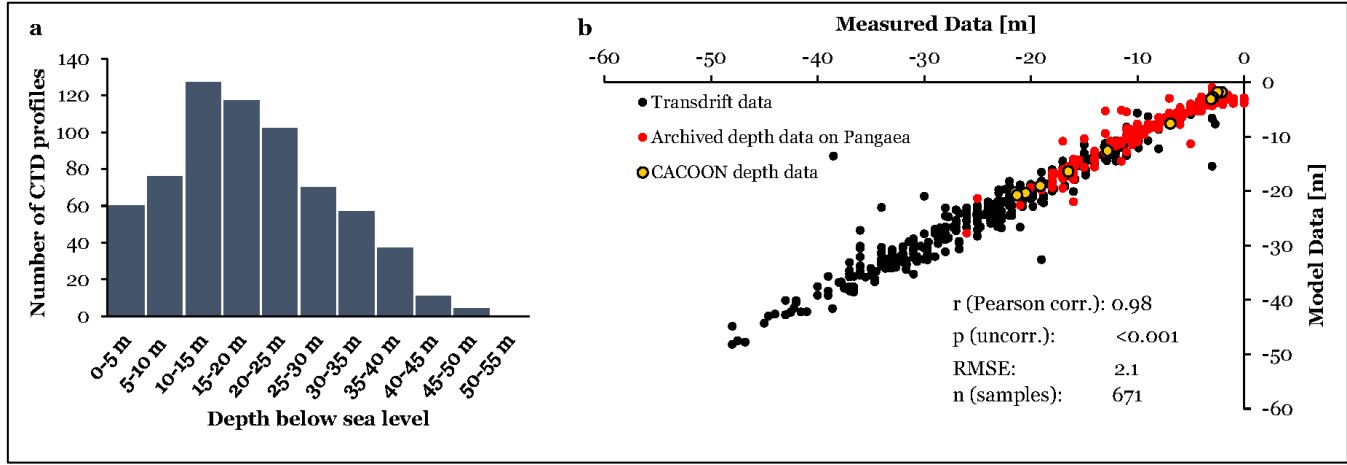

**Figure 6:** Validation data for the topo to raster model (50 m) of the Lena Delta region. a) Histogram showing the depth distribution (5 m depth intervals) of the CTD data in the Lena Delta region. b) Correlation between the CTD data (671 points) and the modelled values ($TTR_{50}$).

## 4.2 Validation of the Kolyma Gulf region bathymetry

The validation of the Kolyma Gulf region bathymetry ($TTR_{50}$) with own depth data (62 measured points, Palmtag and Mann, 2021) showed a very good agreement (Pearson´s correlation is 0.90, $p < 0.001$) (Figure 7). The cross-validation with 1,030 randomly chosen points showed an excellent agreement as well with a Pearson´s correlation of 0.98, $p < 0.001$. Therefore, the Kolyma bathymetry model shows good results. Only the deepest points (deeper than 32 m) are not well represented in the final model output, since they are randomly distributed in the Kolyma channel or along the border of the modelled extent

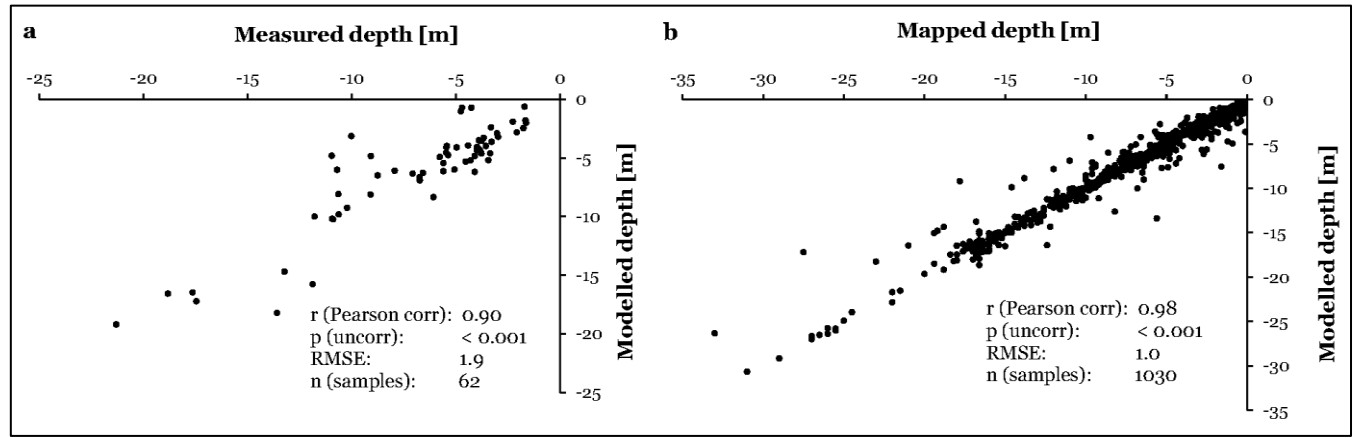

**Figure 7:** a) correlation of the CTD data with the modelled bathymetrical values (TTR$_{50}$) from the Kolyma Gulf region (62 points). b) cross-validation of the modelled bathymetrical values by a subset of samples (1,030), which were excluded from the model run.

### 4.3 Improved representation of near-shore zones compared to the IBCAO

For the comparison of the TTR$_{200}$ with the IBCAO v4 bathymetry (Jakobsson et al., 2020), only those areas where both data sets had values below sea level were included, since the IBCAO v4 does not have depth indications for some near-shore and river channel areas. The comparison for both, the Lena Delta region bathymetry (Figure 8) and the Kolyma Gulf region bathymetry (Figure 9), showed a good agreement. Overall, the mean difference between TTR$_{200}$ and IBCAO is -0.2 ± 1.7 m for the Lena Delta region and -0.2 ± 1.0 m for the Kolyma Gulf region indicating that the TTR$_{200}$ bathymetry slightly overestimates the depths compared to the IBCAO v4 bathymetry. However, the small difference mean value close to zero covers up the major differences between the two data sets. In particular, in close proximity to the coast the spread between the two data sets is bigger (maximum difference of -33.7 m and -18.5 m for the Lena and Kolyma region, respectively). This difference then becomes smaller with larger distance from the shorelines (see appendix Figure F1). This demonstrates, the new high-resolution TTR bathymetry data sets are able to capture small-scale variability close to shore. In addition, with the TTR bathymetry, we are able to identify the deeper parts of the Kolyma river channels and the continuation of the larger channels in the transition from the river mouth to offshore areas for both regions, the Lena Delta (Figure 8a + 8b) and Kolyma Gulf (Figure 9a + 9b). These areas are important for the estimation and the modelling of the river outflow or the nutrient flux output into the coastal zone but are largely missed or underrepresented by the IBCAO v4 (Figure 8c, 8d, 9e). This is a major improvement provided by our new high-resolution bathymetry and highlights the benefit of our TTR bathymetry data sets. In addition, the TTR bathymetry data sets give a more precise depth and water extent estimation for the Kolyma river channels (Figure 9c + 9d) and can depict smaller scale variations in the topography (Figure 9e + 9f).

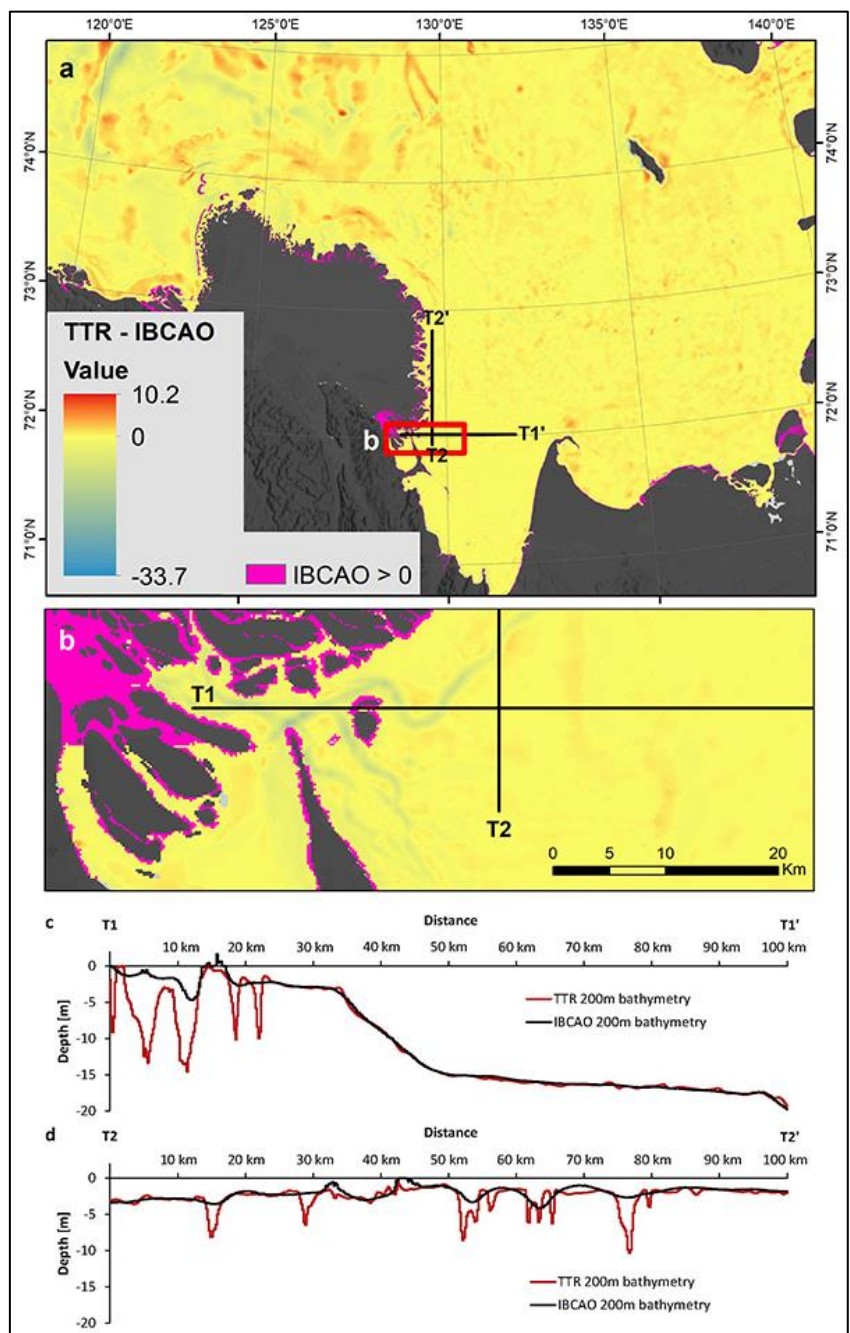

**Figure 8:** Depth difference map for the $TTR_{200}$ bathymetrical model and the IBCAO v4 200 m bathymetrical data set in the Lena Delta region ($TTR_{200}$ minus IBCAO v4). Purple zones show areas where the IBCAO v4 incorrectly provides values above sea level. Reddish areas depict zones where the IBCAO v4 overestimates the depth in comparison to the $TTR_{200}$. Blueish areas show zones where the IBCAO v4 has lower water depths than the $TTR_{200}$. The highest positive value (10.2 m, red) means that the IBCAO v4 indicates a depth, which is 10.2 m deeper than in the $TTR_{200}$. Panel c) and d) show cross-sections deriving depth data from both data sets ($TTR_{200}$ in red and IBCAO v4 in black) in the near-shore zone indicating the higher spatial variability and detection of deep river channels in the $TTR_{200}$.

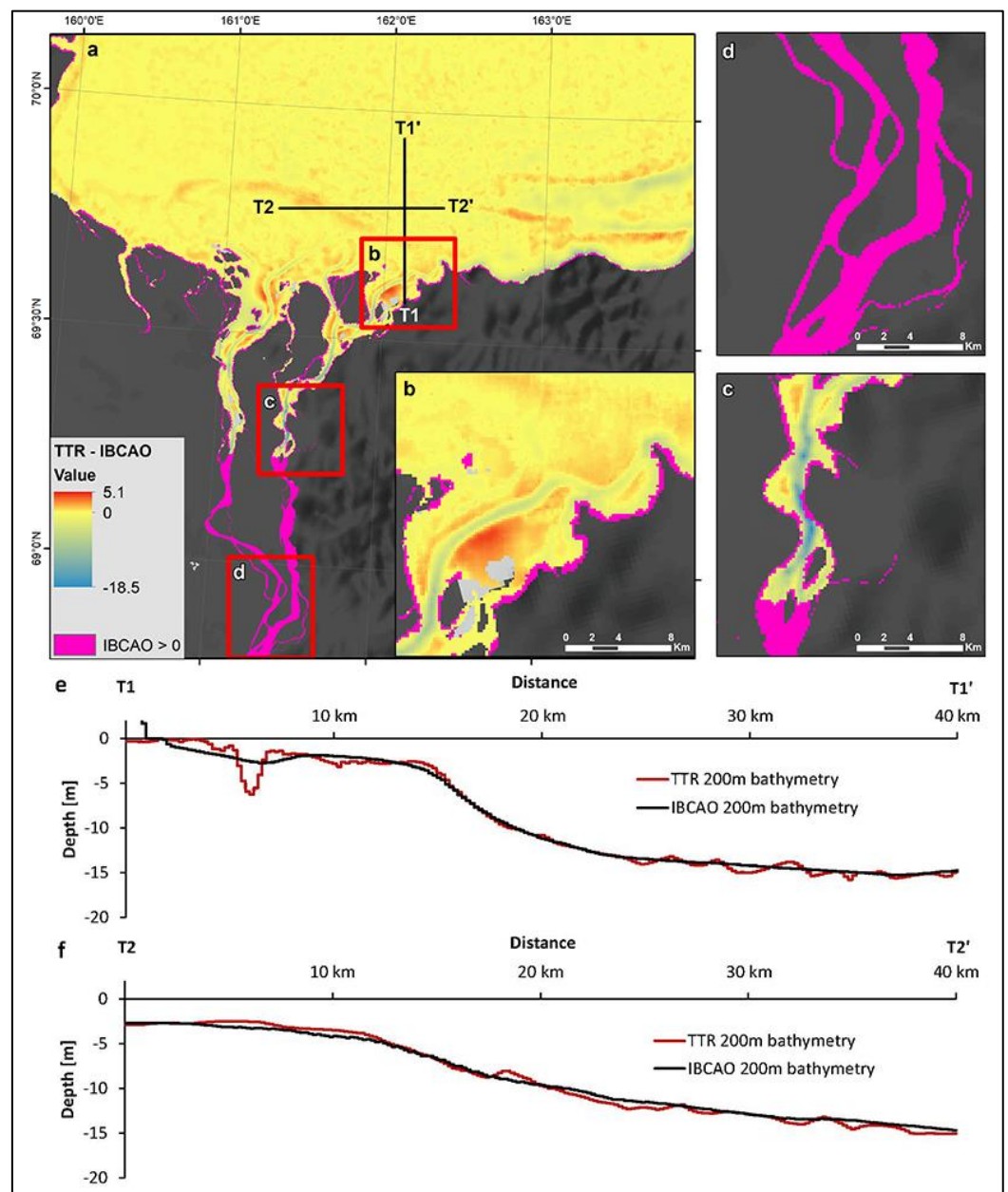

**Figure 9:** Depth difference map for the $TTR_{200}$ bathymetrical model and the IBCAO v4 200 m bathymetrical data set in the Kolyma Gulf region ($TTR_{200}$ minus IBCAO v4). Purple zones show areas where the IBCAO v4 incorrectly provides values above sea level. Reddish areas depict zones where the IBCAO v4 overestimates the depth in comparison to the $TTR_{200}$. Blueish areas show zones where the IBCAO v4 has lower water depths than the $TTR_{200}$. The highest positive value (5.1 m, red) means that the IBCAO v4 indicates a depth, which is 5.1 m deeper than in the $TTR_{200}$. Panel e) and f) show cross-sections deriving depth data from both data sets ($TTR_{200}$ in red and IBCAO v4 in black) in the near-shore zone directly comparing the $TTR_{200}$ and then IBCAO v4 bathymetrical model indicating a generally good agreement between the two data sets with the exception of deep main river channels (e).

## 4.4 Potential applications and usage of the data sets

The resulting $TTR_{50}$ and $TTR_{200}$ products improve the near-shore bathymetry for the Lena Delta and Kolyma Gulf region, correlate closely with available CTD data, and have a close match with the new 200 m resolution IBCAO v4 data set. The reason for this high agreement is certainly the large number of points and isobaths lines included in the $TTR_{200}$ models. However, for near-shore sub-regions the $TTR_{200}$ deviates from the IBCAO v4 data, suggesting a partial lack of spatial detail and higher uncertainty in particular in near-shore zones for the IBCAO v4. It is also important to keep in mind that the input

data for the TTR data sets were collected over a time span from the 1940s to the early 2000s. As a result, some included point depth measurements do not represent the current state anymore. By including water surface layers, which are based on recent satellite data, these point depth measurements were automatically excluded from the model once their location fell outside of the current water area. In particular, in the shallow parts of the river channels, changes could have occurred since the first data collection campaigns. However, Lauzon et al. (2019) reported that Arctic deltas often have persistent river channels due to the

stabilizing effects of ice and permafrost. Also, particularly in the river channels and close to the shore areas, the $TTR_{50}$ and $TTR_{200}$ models show an improvement compared to currently available data products (IBCAO v4) and might help to model river output flow and fluvial carbon exports. In particular, the new, small-scale (local) bathymetry from the two regions is essential for modelling arctic river export in the near-shore in contrast to IBCAO v4, which lacked the spatial variability or the nearshore required for such local scale modelling.

Our detailed coastal zone bathymetry can be used for additional applications such as determining the zone of stable landfast ice. For example, Nghiem et al. (2014) found, by analyzing the recurrence of sea ice fractures, that the landfast ice extent has been stable for decades off the coast of the Mackenzie River Delta and is likely constrained by the near-shore bathymetry. Similar results were observed by Mahoney et al. (2007) who linked the stability of landfast ice to local bathymetry at the Beaufort Sea Coast. The identification of zones of grounding ice in winter can be important, since these zones can enable

a direct heat exchange to the sediment. Our high-resolution bathymetry might also help to understand where seabed scouring by ice can occur, which can cause reworking of bottom sediments and of benthic communities. Reimnitz et al. (1977) found complete reworking of the seabed to an average depth of 20 cm in 6 to 40 m water depths by ice scours and Conlan et al. (1998) showed the change of benthic organisms caused by ice scouring, which they consider to be a large-scale sediment reworking process at Arctic coasts.

In addition, our product can serve as input or validation for other studies. For example, local bathymetry can be used for validating the mapping of landfast sea ice stability with interferometric Synthetic Aperture Radar (Dammann et al., 2019). Further, coastal bathymetry and water level are important parameters for determining rates of coastal erosion (e.g. Barnhart et al., 2014; Pearson et al., 2016) and an improvement of local bathymetry will therefore help to improve model forecasts of coastal erosion in the Laptev Sea region (e.g. Rolph et al., 2021). In addition, for determining the extent of subsea permafrost,

knowledge on water depth and sea surface morphology is essential. High-resolution bathymetry products may improve the delineation and depth estimation of subsea permafrost (Nicolsky et al., 2012; Overduin et al., 2019).

In summary, this is, to our best knowledge, the best available digital bathymetrical data set for the near-shore of the Lena Delta and Kolyma Gulf regions, despite the fact that it is partly based on historical data. The accuracy and resolution of the two data sets provide important spatial information about the depth distribution in the coastal zones of the Lena Delta and Kolyma Gulf regions, and give an indication of the offshore continuation of the main river channels. In combination with detailed mapping of deep channels in Arctic river deltas (Juhls et al., 2021), our new data set will improve the modelling of freshwater pathways as it transitions from land to ocean.

Our main purpose for compiling this data set was to create a digital bathymetrical data set as a model input for a high-resolution implementation of the coupled FVCOM-Arctic European Regional Shelf Seas Ecosystem (Arctic-ERSEM) Model. Additionally, this data set can improve estimates of water and particulate and dissolved matter loads and distribution from the Lena and Kolyma main channels, as well as aid in understanding the dynamics of nearshore landfast ice, ice scouring and bottom sediment disturbance probability, as well as subsea permafrost presence.

## 4.5 Challenges and limitations of the bathymetry models

According to the nautical chart legends, depth measurements have previously been corrected to mean sea level to account for the tidal influence during measurements. Therefore, the influence of astronomical tides, which is small in these areas (less than 1.5 m) (Are and Reimnitz, 2000; Pivovarov et al., 2005), can be neglected for the bathymetrical calculations and a correction based on tidal charts would likely introduce more errors and the benefit would be small. This, however, has to be considered when analyzing the CTD data (see chapter 2.3) in more detail, since no correction has been applied to our own CTD data we collected here.

While tidal influence in these regions is rather small, the coastal zones are affected by highly variable near-shore and river water levels. Wind and particularly storm surges can lead to increased coastal water depths and flooding. In particular, surges can lead to a shift of flow direction in e.g. the Kolyma River mouth area. Water level can be up to 2.5 m higher and lead to a surge, which extends more than 200 km into the Kolyma River (Nikanorov et al. 2011). Similarly, in the Lena Delta, Are and Reimnitz (2000) report that storm surges can cause a water level increase by up to 3.5 m in the northwestern part of the Lena Delta. In addition, the variable river discharge itself can considerably change the water level in the river mouth area of the Kolyma River and can be up to 6 m higher during peak (freshet) runoff periods (Nikanorov et al. 2011). These events and dynamics are not captured in the $TTR_{50}$ and $TTR_{200}$ bathymetry data.

Moreover, the inland continuation of the bathymetry in the Lena River main channels would further improve estimations on river and sediment outflow into the Laptev Sea. However, available maps did not cover these areas and as such no statement on water depths in the main Lena River channels can be made with our bathymetrical data set. Also, Juhls et al. (2021) found several smaller channel continuations offshore the Lena Delta which are undetected even with our high resolution bathymetry.

When using the $TTR_{50}$ and $TTR_{200}$ bathymetry data sets, it is important to consider that these data sets are based on spatial interpolation. As many other spatial interpolation techniques, topo to raster assigns values to cells based on the

surrounding data points. The advantage of topo to raster is that it also includes lines (isobaths) to model the surface accurately. The $TTR_{50}$ and $TTR_{200}$ are products of such an interpolation, where the point density varies (see appendix figure D1 + D2) and the median distance between points is 238 m and 794 m for the Kolyma Gulf and Lena Delta region, respectively. The average point density in the TTR models is 0.2 points/sqkm for the Lena Delta (figure D1) and 1.5 points/sqkm (figure D2) for the Kolyma Gulf region. However, the topo to raster method does also include the isobaths, which are particularly important for the near-shore zone and helped to improve the accuracy in depicting the near-shore river channels. In addition, the point density is considerable higher in near-shore areas compared to further offshore zones, particularly for the Lena Delta region (see appendix figure D1).

## 5 Disclaimer

It is important to note that this data set is based on historical data and therefore deviations from the present state might occur. In particular, river channels depths may vary over seasonal timescales. Therefore, this product is an approximation and a best currently available attempt to map coastal zone and river delta/gulf bathymetry in this understudied yet rapidly changing region. Additional products are needed in the future, particularly from upstream river channels to determine and map the extent and stability of the shorelines as well as the area of dynamically changing zones (e.g. sandbanks).

Both digital products, the Lena Delta region bathymetry and the Kolyma Gulf region bathymetry were not designed and tested for navigational purposes and should therefore not be used as navigational guidance. These data sets were established only for scientific purposes and lack the required actuality for navigation and shipping today.

## 6 Data availability

The two bathymetrical data sets ($TTR_{50+200}$) for both the Lena Delta and Kolyma Gulf region are available on the PANGAEA (www.pangaea.de) data set repository as GeoTiffs in 50 m ($TTR_{50}$) and 200 m ($TTR_{200}$) spatial resolution (https://doi.pangaea.de/10.1594/PANGAEA.934045 (Fuchs et al. 2021a) for the Lena Delta and https://doi.pangaea.de/10.1594/PANGAEA.934049 (Fuchs et al., 2021b) for the Kolyma Gulf region. In addition, the published data sets include the depth point input data, the isobaths line input data and the water area polygon input data for both, the Lena Delta and the Kolyma Gulf region in shapefile format.

The CTD data from the CACOON expeditions are archived on PANGAEA for the Lena Delta region (https://doi.pangaea.de/10.1594/PANGAEA.933187) (Fuchs et al. 2021c) and on the BODC for the Kolyma Gulf region (https://doi.org/10.5285/c10a2798-40cc-7648-e053-6c86abc07c3c) (Palmtag and Mann, 2021) data set repository, respectively.

## 7 Conclusions

With our new data set we provide the first detailed and seamless digital models of coastal zone bathymetry for the Lena River
Delta and Kolyma Gulf region in north-eastern Siberia. We provide geotiff rasters in 50 m and 200 m spatial resolution based
on digitized depth points and isobaths lines from nautical charts. The models were compared to measured coastal zone depth
data, archived depth data available on PANGAEA and the IBCAO bathymetry. While the new bathymetrical models showed
a good agreement to the compared data, the new models particularly reveal the location and continuation of the larger, deeper
river channels in the transition from the river mouth to offshore areas for both regions. Our data product can therefore serve
as model input to quantify fluvial and coastal carbon fluxes as it transitions from land to ocean but also help to understand
dynamics of nearshore landfast ice or subsea permafrost in coastal zones.

## Appendix A – Extent of nautical charts

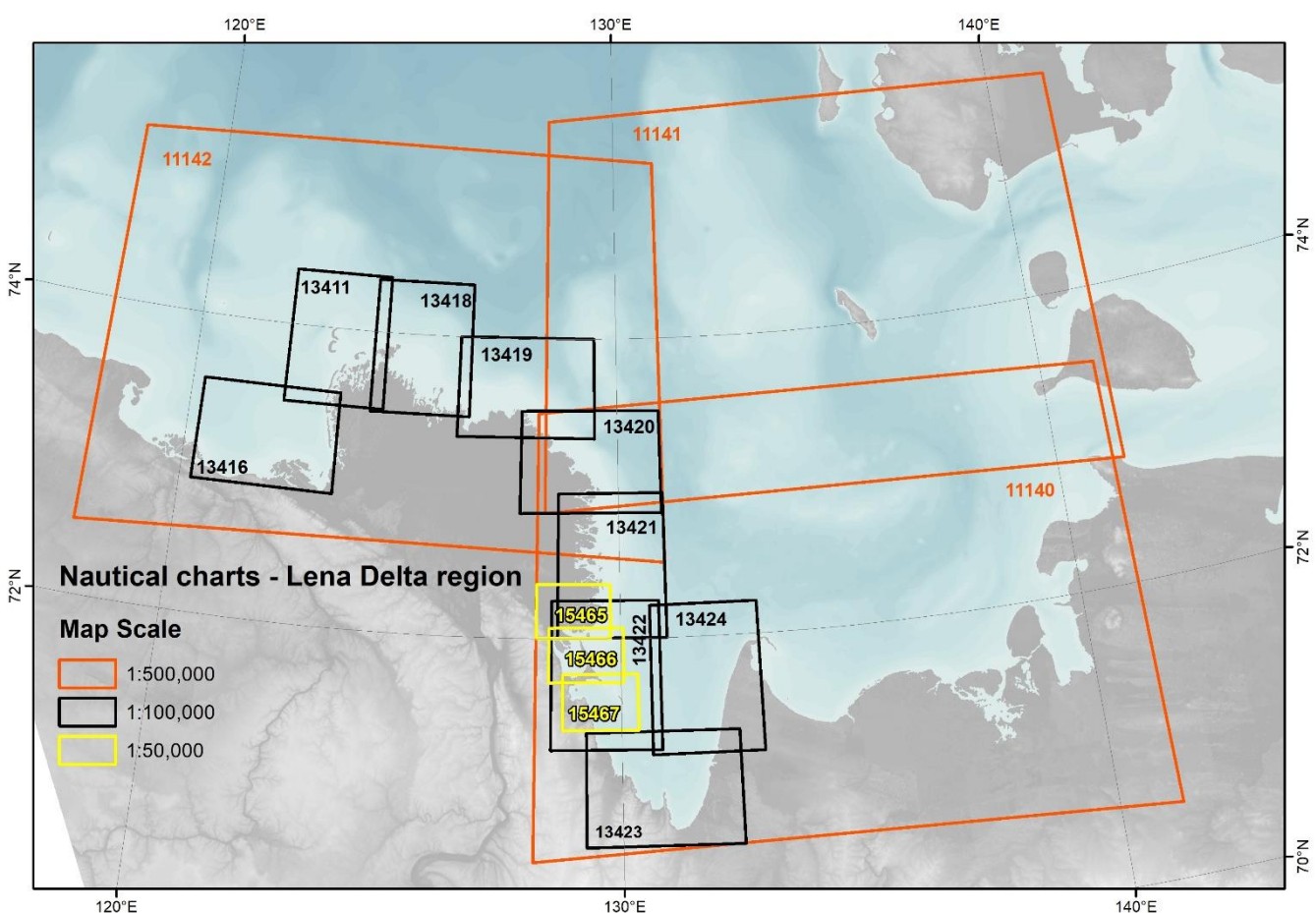

**Figure A1**. Extent and scale of the nautical charts used for the bathymetrical models in the Lena Delta region. Chart numbers are indicated in the map. More details on the charts can be found in Table 1. Background map is the IBCAO v4 data from Jakobsson et al. (2020).

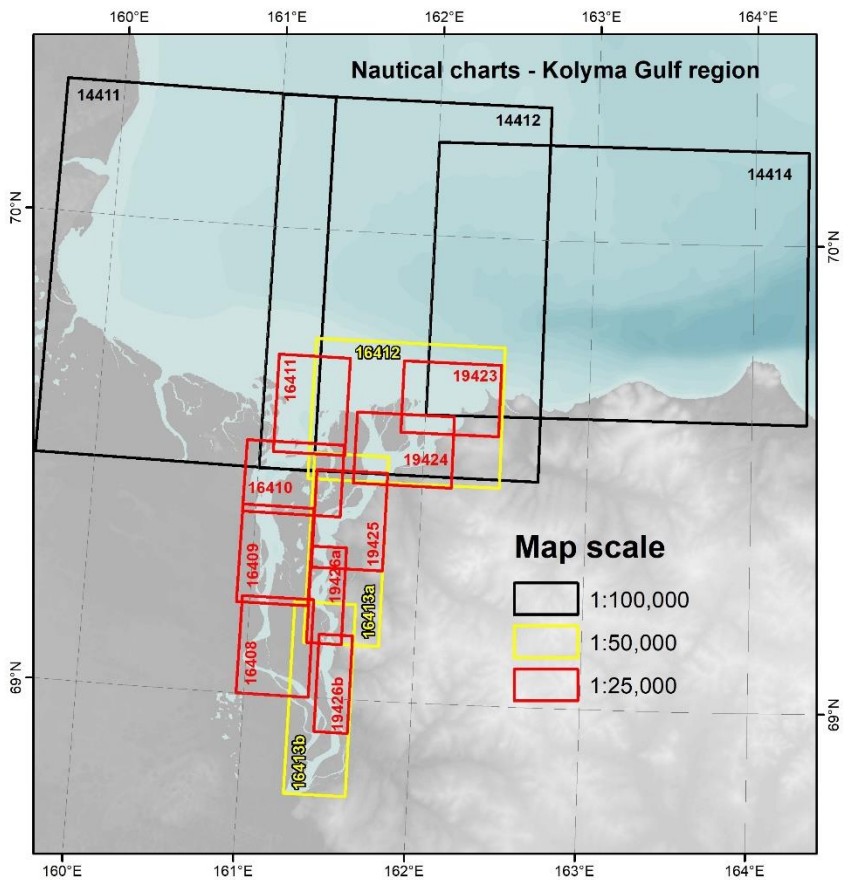

**Figure A2**. Extent and scale of the nautical charts used for the bathymetrical models in the Kolyma Gulf region. Chart numbers are indicated in the map. More details on the charts can be found in Table 2. Background map is the IBCAO v4 data from Jakobsson et al. (2020).

**Appendix B – Water area delineation**

We used the water area to set the boundary for the bathymetry models in the topo to raster tool. In order to get an accurate representation of the water area, a Normalized Difference Water Index (NDWI) (Gao, 1996) was applied to multispectral Landsat-8 imagery for the Lena Delta region. Band 6 and band 3 from twelve cloud free Landsat 8 scenes (Table S1) were

used to calculate the NDWI. The NDWI (band 6 – band 3 / band 6 + band 3) resulted in values between -1 and 1. All values larger than 0.05 were identified as water as this allowed to separate sand banks from water areas.

**Table B1.** Landsat 8 scenes used for calculating the NDWI

| Landsat ID | Acquisition date | Path/Row |
|---|---|---|
| LC81912362018222LGN00 | 2018-08-10 | 191/236 |
| LC81300062019214LGN00 | 2019-08-02 | 130/006 |
| LC81250072018240LGN00 | 2018-08-28 | 125/007 |
| LC81230082020264LGN00 | 2020-09-20 | 123/008 |
| LC81230092020264LGN00 | 2020-09-20 | 123/009 |
| LC81230102020264LGN00 | 2020-09-20 | 123/010 |
| LC81270092018222LGN00 | 2018-08-10 | 127/009 |
| LC81260102020253LGN00 | 2020-09-09 | 126/010 |
| LC81290092016247LGN01 | 2016-09-03 | 129/009 |
| LC81320082018257LGN00 | 2018-09-14 | 132/008 |
| LC81350082020268LGN00 | 2020-09-24 | 135/008 |
| LC81370082019215LGN00 | 2019-08-03 | 137/008 |

For the Kolyma Gulf region the Global surface water layer by Pekel et al. (2016) was used to delineate the water boundary with all areas, which are >90% of the time covered by water were considered as water body. We are aware of wind- and tide-driven water level fluctuations, particularly in the river mouth, but the water occurrence dataset was only used for delineating the outer boundary (maximum normal water extent) of the topo to raster calculations to avoid interpolation outside of measured
points or between points where land areas or islands are located.

**Appendix C – CTD specifications**

During the CACOON 2019 expeditions (Fuchs et al., 2021d) in the Lena Delta region CTD measurements were taken with a handheld SontekTM CastAway sensor with an integrated GPS. The measured data include pressure (dbar; accuracy: 0.25%),
depth (m; ±0.25%), temperature (°C; ±0.05°C), conductivity (mS/cm; 0.25%±5 mS/cm), specific conductance (mS/cm; 0.25%±5 mS/cm), salinity (practical salinity scale; ±0.1), sound velocity (m/s; ±0.15 m/s), and density (kg/m$^3$; ±0.02 kg/m$^3$). In total, 31 depth profiles were measured from the sea (or river) water surface to the sea (or river) bottom (Table B1).

        In the Kolyma Gulf region, CTD measurements were taken using a HYDROLAB HL7 multiparameter probe from small motorboats (Palmtag and Mann, 2021). Following parameters were collected on each cast: specific conductivity (mS/cm;
± (0.5% of reading + 0.001 mS/cm), turbidity (NTU; ± 1%), barometric pressure (mmHg; +/- 1 mmHg), dissolved oxygen

(mg/L; ± 0.2 mg/L), depth (m; ± 0.05m), water temperature (°C; ± 0.10 °C), density (kg/m$^3$; ±0.02 kg/m3), salinity (psu; ±0.1) and chlorophyll-a (µg/L; ± 3%).

**Table C1**. Location and date of the CTD measurements in the Lena Delta region with the handheld SontekTM CastAway sensor

| Site | Latitude | Longitude | Cast date | Depth |
|---|---|---|---|---|
| CAC19-01 | 72.509039 | 129.248017 | 30.03.2019 | 10.58 |
| CAC19-02 | 72.516828 | 129.545513 | 29.03.2019 | 2.46 |
| CAC19-03 | 72.525361 | 129.841995 | 30.03.2019 | 3.14 |
| CAC19-04 | 72.525494 | 129.863887 | 31.03.2019 | 2.88 |
| CAC19-23 | 72.521360 | 129.693019 | 31.03.2019 | 2.07 |
| CAC19-A | 72.501279 | 129.101644 | 01.04.2019 | 11.69 |
| CAC19-B | 72.479381 | 128.971100 | 01.04.2019 | 5.32 |
| CAC19-C | 72.455611 | 128.844519 | 02.04.2019 | 1.92 |
| CAC19-D | 72.461555 | 128.694496 | 02.04.2019 | 17.85 |
| CAC19-E | 72.501903 | 128.629750 | 03.04.2019 | 2.66 |
| CAC19-F | 72.518782 | 128.492189 | 03.04.2019 | 3.25 |
| CAC19-G | 72.535431 | 128.353264 | 04.04.2019 | 8.07 |
| CAC19-H | 72.564080 | 128.238459 | 04.04.2019 | 3.21 |
| CAC19-S-04 | 72.530128 | 130.126304 | 03.08.2019 | 6.87 |
| CAC19-S-05 | 72.539833 | 130.433507 | 03.08.2019 | 12.79 |
| CAC19-S-06 | 72.541183 | 130.722484 | 03.08.2019 | 16.51 |
| CAC19-S-07 | 72.550563 | 131.018370 | 03.08.2019 | 19.13 |
| CAC19-S-08 | 72.554506 | 131.314719 | 03.08.2019 | 20.56 |
| CAC19-S-09 | 72.559000 | 131.606328 | 03.08.2019 | 21.51 |
| CAC19-S-10 | 72.553048 | 131.914890 | 03.08.2019 | 21.35 |
| LEN19-S-01 | 72.399384 | 126.695646 | 09.08.2019 | 18.62 |
| LEN19-S-02 | 72.536958 | 126.928427 | 09.08.2019 | 16.73 |
| LEN19-S-03 | 72.627117 | 127.419353 | 09.08.2019 | 5.59 |
| LEN19-S-04 | 72.633475 | 127.959208 | 09.08.2019 | 2.67 |
| LEN19-S-05 | 72.563824 | 128.244662 | 09.08.2019 | 4.51 |
| LEN19-S-06 | 72.521071 | 128.515459 | 08.08.2019 | 7.74 |
| LEN19-S-07 | 72.461339 | 128.695025 | 08.08.2019 | 16.37 |
| LEN19-S-08 | 72.477073 | 128.970640 | 08.08.2019 | 7.11 |
| LEN19-S-09 | 72.509043 | 129.248415 | 08.08.2019 | 10.27 |
| LEN19-S-78 | 72.452992 | 128.840959 | 08.08.2019 | 9.12 |
| LEN19-S-89 | 72.501724 | 129.097863 | 08.08.2019 | 12.91 |

**Appendix D – Point cloud density maps**

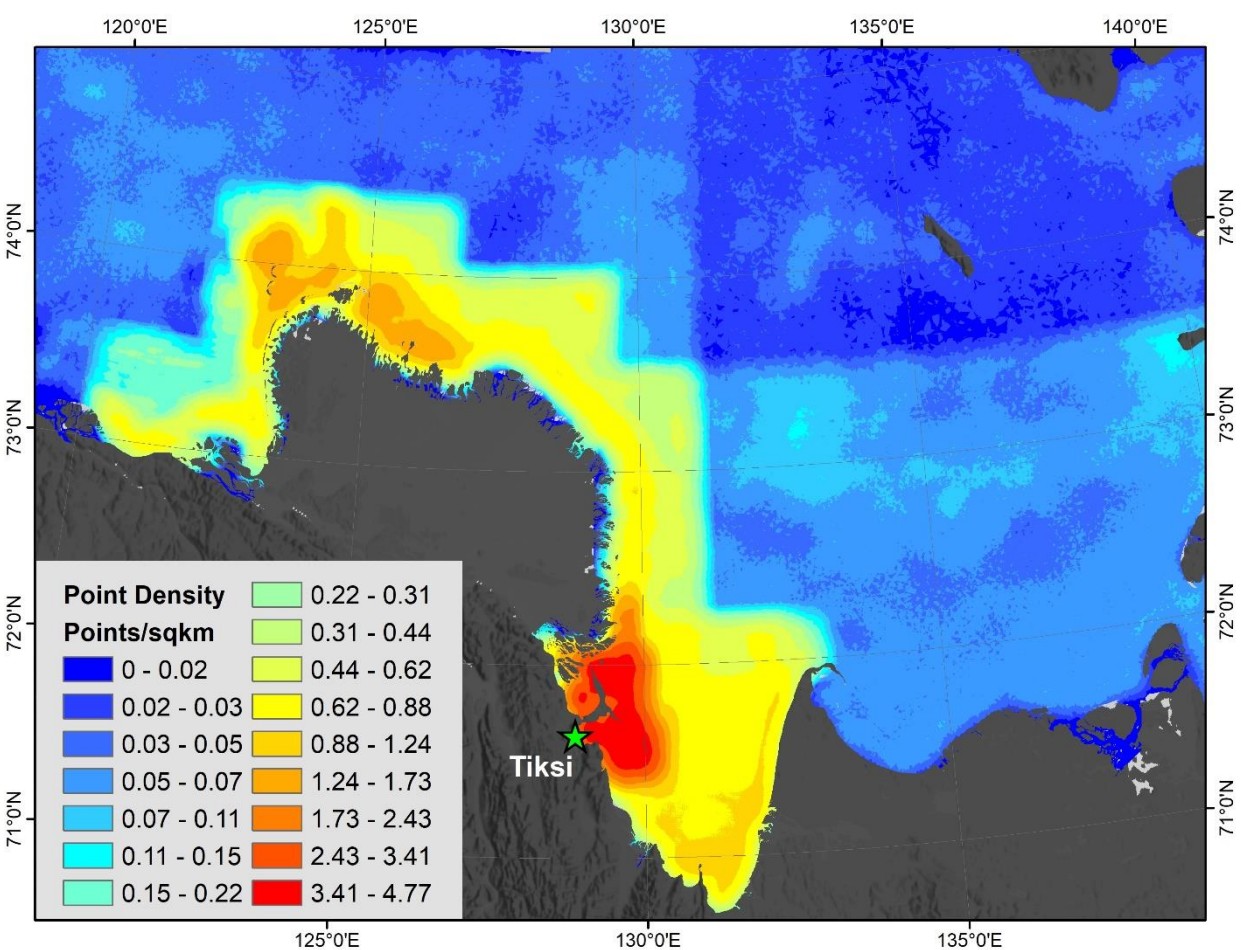

**Figure D1.** Point density for the Lena Delta bathymetrical models ($TTR_{50}$ + $TTR_{200}$) indicating the highest point density in the coastal areas around Bykovsky Peninsula. The green star shows the location of the city and harbour of Tiksi.

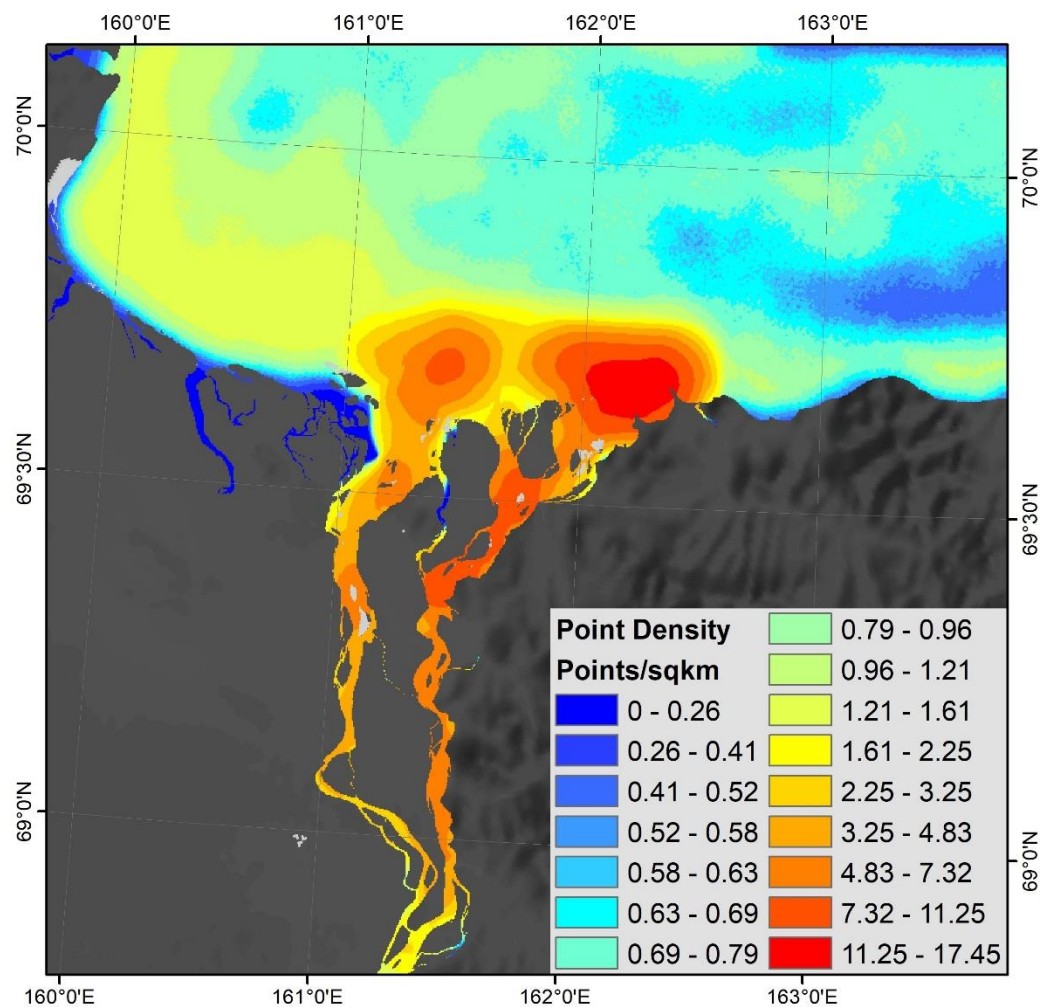

**Figure D2.** Point density for Kolyma Gulf bathymetrical models ($TTR_{50}$ + $TTR_{200}$) indicating the higher point density in the Kolyma main channels.

 **Appendix E – Validation points deviating from the Topo to raster model**

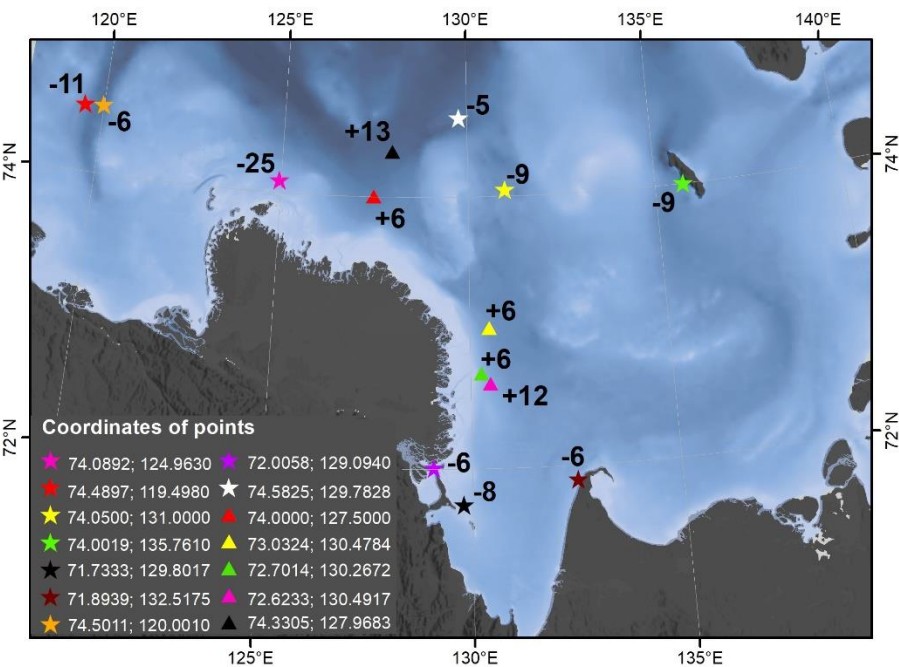

**Figure E1.** Validation points in the Lena Delta region, which deviate by more than 4.3 m (difference larger than two standard deviations from the mean distance) from the topo to raster (50m) model. Negative values (stars) indicate that the validation point is x meters deeper than the depth indicated by the $TTR_{50}$. Positive values (triangles) indicate that the validation points report a shallower depth by x meters compared to the $TTR_{50}$ model. The coordinates of the points are given in the map legend.

**Appendix F – Topo to raster and IBCAO v4 comparisons**

The $TTR_{200}$ bathymetrical models were compared to the IBCAO v 4 model (Jakobsson et al., 2020). Overall difference between the two data set was small with a mean difference of -0.2 ± 1.7 m for the Lena Delta region and -0.2 ± 1.0 m for the Kolyma Gulf region indicating that the $TTR_{200}$ bathymetry slightly overestimates the depths compared to the IBCAO v4 bathymetry. However, when splitting the maps into zones with a certain distance from the coast, distinct differences become visible (Figure F1). In particular, the difference and the spread are large in near-shore zones (0-10 km from the shorelines), whereas the variance of the difference between the two products becomes smaller with larger distance from the coast for the Kolyma Gulf region. For the Lena Delta region, a similar picture can be observed, however, in the areas >30 km away from the coast, the differences between the two products increases again.

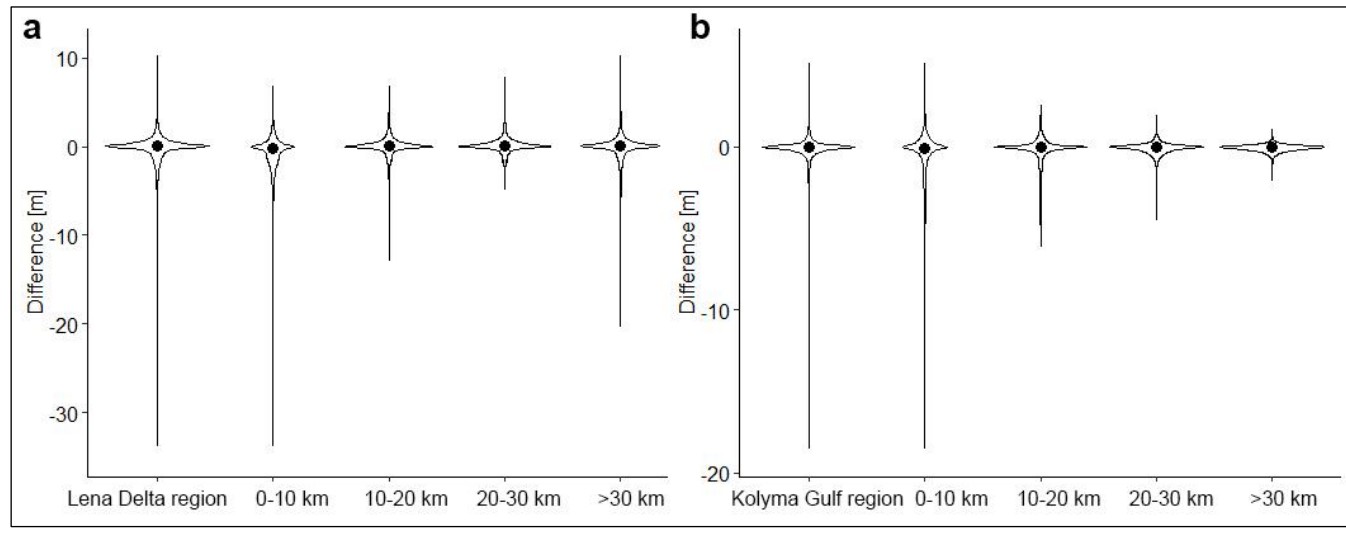

**Figure F1.** The difference of the topo to raster (TTR$_{200}$) and IBCAO v4 bathymetry (Jakobsson et al., 2020) shown in relation to distance from the shore in violin plots. The plots are based on the difference (TTR$_{200}$ – IBCAO v4) layers where negative values indicate that the IBCAO v4 underestimates the depth compared to the TTR$_{200}$ and positive values indicate an overestimation of the depth by the IBCAO v4 in comparison to the TTR$_{200}$ bathymetry. The y-axis shows the absolute
difference between the two bathymetrical products. Black dots show the median value. Areas where the IBCAO v4 bathymetry shows values above mean sea level were excluded from this analysis (see purple areas in Figure 8 and Figure 9 in the main text). a) shows plots of the Lena Delta region. The first plot to the left includes the entire Lena Delta region, whereas the other four plots show areas in certain distances from the coast (0-10 km, 10-20 km, 20-30 km, and >30 km away from the shore). b) shows plots of the Kolyma Gulf region. The first plot to the left includes the entire Kolyma Gulf region, whereas the other
plots represent areas with certain distances from the coast (0-10 km, 10-20 km, 20-30 km, and >30 km away from the shore). Please note the different scales of the y-axes. The TTR$_{200}$-IBCAO v4 difference shows that particularly in the coastal zones and near-shore areas the IBCAO underestimates the depth by not capturing the small-scale variability. In contrast, the TTR$_{200}$ bathymetry detects the continuation of the Lena and Kolyma main channels from the river mouth into offshore regions.

**Funding**

This study was funded by the NERC-BMBF project CACOON (NE/R012806/1 and #03F0806A - Changing Arctic Ocean (CAO) programme https://www.changing-arctic-ocean.ac.uk/project/cacoon/)

**Author contribution**

MF, JS, GG and PO designed the concept of the study. AA, MF, and PO digitized nautical maps. MF applied the raster calculations in ArcGIS 10.6. MF, JP, OO, TS, and JS carried out field work and collected CTD measurements. MB set the
485 requirements for and tested the suitability of the data set. MF wrote the initial draft of the manuscript. All authors contributed to the writing and editing of the manuscript.

**Competing Interests**

The authors declare that they have no conflict of interest.

**Acknowledgements**

We thank the Hydrobase Tiksi and the Lena Delta Reserve for the support with field logistics during the Spring 2019 field campaign and we are grateful for the support by the Samoylov Research Station for the fieldwork in Summer 2019, which took place in the framework of the joint Russian-German Expedition *Lena 2019*. We thank the Cherskiy Northeast Science Station staff for support of the Kolyma fieldwork in summer 2019 and we are grateful for the hospitality of the ship crews from the vessels *Anatoliy Zhilinksiy* and *Merzlotoved*. We thank Sofia Antonova for transliterating the maps.

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
