# Peer review of "High-resolution bathymetry models for the Lena Delta and Kolyma Gulf coastal zones"

_Earth System Science Data, 2021_

## Author Response (AR1)

**Reply to anonymous reviewer #1**

**Reviewer comment (RC):** The manuscript presents the 50 m and 200 m resolution bathymetrical models for the Lena Delta and Kolyma Gulf regions based on digitized nautical charts. The authors provide detailed information on the generation of the models and its validation. Additionally, they discuss the comparison with existed data and limitations of the models as well as potential applications and usage of the data sets. The first detailed and seamless digital high resolution models is the best available digital bathymetrical data set with the high accuracy and resolution and therefore is very valuable contribution for Arctic studies. Detailed bathymetry model is necessary for better quantification of fluvial and coastal carbon fluxes to the Arctic Ocean as well as for other studies related to Arctic delta and near-shore dynamics. Great advantage of the models is using field measurements conducted by authors which show a strong correlation between model and field data. Also additional existed data for model validation were used. High resolution of the bathymetry models allows to reveal deeper parts of the Kolyma and Lena Delta river channels and the transition and continuation of the main channels into the near-shore and deeper coastal areas. Additional benefit of the models is the coverage of the coastal near-shore zone which had sparse coverage in other data sets.
The manuscript is well-written and well-structured with good and clearly presented figures and tables. I would consider it to publish in ESSD after a minor revision.

**Authors reply (AR):** We kindly thank the anonymous reviewer for this very positive and helpful feedback for our study and hope to address all the points raised by the reviewer in our reply below.

**RC**: My general comments are:
1. I recommend for better perception to combine 2.3-2.5 to one section like "3.3 Model validation..." with subsections 3.3.1 Field measurements, 3.3.2. Existed archive data and 3.3.3. Comparison to IBCAO.
2. It is not critical but would be great to add the figure to appendix with coverage of the nautical charts as they have different scale. Also it is not clear which areas are covered by maps of which scale. Or all study area covered by maps of scale from 1:25,000-1:100,000? Then it should be noted in text.
3. Some section titles better to name more clear which I'll note in listed below detailed comments.

**AR:**
1. Thank you for your suggestions, we merged chapter 2.3-2.5 to "2.3 Model validation and comparison to existing bathymetry products" and included subchapters as suggested.
2. We added additional maps to the appendix (Figure A1+A2) with the outline of the nautical charts, including the scales. In addition, we included a reference to this figure in in chapter 2.1.
3. Thank you. We followed this suggestion and changed the headlines accordingly.

**RC:** Detailed comments and suggestions are listed below:
Abstract
22 to add scale of used nautical maps
**AR:** We added the scales (1:25,000 – 1:500,000) of the nautical maps to the abstract.

**RC:** 23 to add the resolution of created models
**AR:** We added the resolution of our models to the abstract.

**RC:** Introduction
40 to add permafrost temperature rising

**AR:** We changed the sentence to "… *climate change-induced increase of permafrost temperatures*…"

**RC:** 62 The usage of "region" should be uniform in the text, while "Region" or "region" are used. I would use "region" with lower case as there is no formal names of Lena Delta or Kolyma Gulf regions.

**AR:** We agree with the reviewer and spelled region with a lower-case R throughout the manuscript.

**RC:** 64 What are this models - are they planned or they are already existed? Add reference if it exists.

**AR:** This model deals with ecosystem simulations from shelf seas to the global ocean and was started to model the lower trophic levels of the marine food web (Butenschön et al. 2016). The arctic ERSEM is currently developed and is an extension to the regular ESRM by including specific Arctic parameters such as permafrost thaw and dissolved organic matter output from rivers. The aim is a better understanding how dissolved organic matter input affects the ecosystems in shelf areas in the Arctic (Bedington et al. 2021.)
We added the references in the manuscript and wrote that the model is in planning.

**RC:** Material and Methods
86 Double usage of "input data". Maybe to use the "primary" instead one of them.

**AR:** We replaced the first "input data" with "primary".

**RC:** 88 to add the depth of near-shore zone

**AR:** We added the depth.

**RC:** 89 for which region?

**AR:** We added the following part: "… *for the coastal zones of the Indian Ocean.*"

**RC:** 90 Different scale maps were overlapped or there were regions with only of one scale map existence?

**AR:** Yes, there are areas, which are covered by only one nautical chart. We produced additional figures (Fig. A1 + A2) showing the extent of the nautical charts, from which it will become more clear which area is covered by which chart and what the original map scale is. In addition, we have figures D1 and D2, which show the point density for each region.

**RC:** 105 I would change the section name to smth like "Creation of bathymetry model based on..."

**AR:** We changed the section name into: Creation of the bathymetrical models based on the Topo to Raster interpolation method.

**RC:** 122-123 Add to abstract

**AR:** We added the spatial resolution of the models (50 m and 200 m) to the abstract.

**RC:** 130 Not clear section name. I recommend to change it on "Field measurements from the near-shore" or similar.

**AR:** We changed it according to your suggestion.

**RC:** 160 Fig. 2 To add the date of measurements (as for Kolyma Gulf region)
**AR:** Adding the date to the map 2a would make the map hard to read due to too much text, since the measurements were taken on a different days. Instead, we added the date range of the measurements in the figure caption and we added a separate table to the appendix (Table B1) with the date and the depth for each of the measurements for the Lena Delta region. We added a reference for this table in the figure caption of figure 2.

**RC:** 165 I recommend to name this section as "Additional Pangaea archive data for model validation" or similar.
**AR:** We renamed the headline to "Additional archived data for model validation".

**RC:** 180 Fig.3 Add "from PANGAEA archive data". Red circles are not well seen, better to change to more contrast color.
**AR:** We increased the contrast in the figure and added PANGAEA to the map key.

**RC:** 181 I recommend to change the section name to " Validation of model in comparison with IBCAO" or similar
**AR:** Since we do not validated but only compared our models to the IBCAO, we changed the headline into: "Comparison of the bathymetrical models to the IBCAO".

**RC:** Validation, comparison, and limitations
258 4.3 I recommend again to change the section name to " Validation of model in comparison with IBCAO" or similar
**AR:** We changed the headline into: "Improved representation of near-shore zones compared to the IBCAO".

**RC:** 259-260 second "comparison" is not necessary
**AR:** Thank you. We removed the second "comparison".

**RC:** 338 It is not clear what does mean the "correction to mean sea level" of depth measurements, please explain it
**AR:** The depth measurements for the nautical charts have been corrected to respect the tidal stage during the measurements in order to avoid over- or underestimation of the true depth. We added the following sentence in the text to make that more clear: "According to the nautical chart legends, depth measurements have previously been corrected to mean sea level to account for the tidal influence during measurements."

**RC:** 341 Which are the heights of such tides?
**AR:** The tides are lower than 1.5 m. We added this information in the text.

**RC:** 348 Delta?
**AR:** We specified to "Lena Delta".

**RC:** 349 Seems that "up to 5" is an extra
**AR:** We deleted "up to 5".

**RC:** 351 Double usage of "further". First maybe to change to moreover
**AR:** We changed the first "further" to "moreover".

**RC:** 361 Add the reference to C1 and C2 figures

**AR:** We added the references to the figures in the text.

**RC:** 384 To add to conclusion the advantages of models such as revealing deeper parts of the river channels and the transition and continuation of the main channels into the near-shore and deeper coastal areas as well as coverage of the coastal near-shore zone which had sparse coverage in other data sets.

**AR:** Thank you for these suggestions. We added this information to the conclusions.

**RC:** 387 Field-measured, PANGAEA archive data and IBCAO...

**AR:** We added "archived depth data available on PANGAEA" to the conclusions.

We are thankful for the valuable comments and suggestions by reviewer #1, which helped to improve our manuscript.

References:

Bedington, M., Torres, R., Polimene, L., Wallhead, P., Juhls, B., Palmtag, J., Strauss, J., and Mann, P. J.: Impacts of riverine terrestrial organic matter on the lower trophic levels of an Arctic shelf ecosystem, *Abstract EGU General Assembly 2021*, https://doi.org/10.5194/egusphere-egu21-12897, 2021.

Butenschön, M., Clark, J., Aldridge, J. N., Allen, J. I., Artioli, Y., Blackford, J., Bruggeman, J., Cazenave, P., Ciavatta, S., Kay, S., Lessin, G., van Leeuwn, S., van der Molen, J., de Mora, L., Polimene, L., Sailley, S., Stephens, N., and Torres, R.: ERSEM 15.06: a generic model for marine biogeochemistry and the ecosystem dynamics of the lower trophic levels, *Geosci. Model Dev.*, 9, 1293-1393, https://doi.org/10.5194/gmd-9-1293-2016, 2016
* * *
**Reply to anonymous reviewer #2**

**Reviewer comment (RC)**: In their manuscript "High-resolution bathymetry models for the Lena Delta and Kolyma Gulf coastal zones", Fuchs et al. provide bathymetric models of 50 m and 200 m resolution of the Lena Delta and Kolyma Gulf coastal zones. The data underlying the models is based on nautical charts published since the 1940's. The authors provide a comparison with recent depth measurements independent of the nautical charts, showing a very good agreement. The authors make it very convincing that their bathymetry is a significant step forwards for these poorly surveyed region, and especially the near coastal zone.

I find little to criticize with this paper. It is well structured, easy to follow with well annotated figures. The Pangaea-download contents are easy to load and understand in GIS. Congratulations to the authors for their work.

**Author´s reply (AR)**: We are grateful and thank reviewer #2 for this very positive feedback and are happy about the feedback that our data set is easily accessible and usable.

We hope to address with this reply all the comments and questions raised by the reviewer.

**RC:** I have a few suggestions, which are however not critical for acceptance in Earth-System Science Data (were I to decide).

**AR:** Thank you for your suggestions. We hope to answer all your questions with this reply.

**RC:** Abstract: I am missing a mention of the resolution of the available models here. Also, it is not clear to the reader of the abstract what "large-scale" nautical maps are.

**AR**: Thank you for this comment. We added the scale of the nautical maps used (1:25,000 – 1:500,000) and the resolution of the final bathymetrical models (50 m and 200 m) to the abstract.

**RC**: 95ff The difference could also be related to the decades inbetween the measurements of the map. This comes up in the discussion later on, but the thought could be introduced here as well. I agree with the procedure using the higher resolution maps though. In this regard, it would be a suggestion to provide maps showing the boundaries and overlapping areas of the nautical charts. It appears the boundaris could be easily added to Figure C1 and C2 in the supplement (they "shine through" in the point densitis anyway). A further suggestion would be to add the date of the nautical chart (or reference to the nautical chart) as an attribute to the shape file. It may be relevant for future studies to have the age of the depth measurement. I fully realize the effort to do so may be prohibitive and this cannot be done.

**AR**: Yes, we agree, the difference between the maps certainly can have several reasons, among them the different times of the map production. We added a short note in the text about the different survey times and different map compilation dates. In addition, we added figures in the appendix of the revised manuscript showing the extents of the nautical charts (Figure A1 for the Lena and Figure A2 for the Kolyma region). Also, we added an additional column to the attribute table of the depth points indicating a reference to the nautical chart. However, we cannot add a year of collection to each point because the nautical charts consist of points collected during multiple years. There unfortunately is no information on the nautical charts specifying in which year an individual depth point was measured.

**RC:** 170 Is it certain that the CTD profiles were cast down to the seafloor?

**AR:** For our own collected data points, we are confident that we reached the seafloor, because we added an additional weight to the CTD device and had the ships sonar as a rough benchmark for the depth at the sampling location. For the Transdrift data we are confident that the depth measurements are very accurate too. Janout et al. (2017) writes that in shallower waters (<200m), the water column was profiled all the way to the seafloor, while in deeper waters, only the upper 200-350m were sampled. Since we did not use sample points deeper than 200 m we are confident that the validation points reached the seafloor.

**RC:** 229: It could be argued to integrate chapter 4.1 and 4.2 to chapter 3, although this is probably a matter of taste.

**AR:** Thank you for this suggestion. Our idea behind that was to only have the models and results in chapter 3 and validation of the results in chapter 4 as a discussion. In that case, we have a results and discussion chapter.

**RC:** 244: I do not agree with this argumentation. The outliers deviate by partly more than 5 m up to almost 20 m. If these were real bathymetric features, it would be worthwhile to plot their location somewhere.

**AR**: Thank you for this comment. Yes we agree, the labelling of these points as "outlier" is not correct. These points are depth measurements at locations where our model does not perform well. We added an additional figure in the Appendix (Appendix E, Figure E1) showing the location of these 14 points including the deviation from our model. As a threshold, we selected all points which deviate by more than two standard deviations from the mean (> 4.3 m) deviation. In addition, we changed the sentence in the main text to:

"A few validation points show a larger deviation from the model (> 5m). These points may indicate real bathymetric features such as small scale variabilities in the sea floor, which are not captured by $TTR_{50}$ bathymetry. The location of these points including the deviation from the $TTR_{50}$ bathymetry are presented in the appendix E (figure E1)."

**RC:** 339ff: What would the values of the astronomical tides be?
**AR:** We added an additional reference with the information that the tidal range is less than 1.5 m.

We thank reviewer #2 for the constructive comments, which helped to improve our manuscript and hope we addressed all the questions raised by the reviewer.

References:
Janout, M. A., Hölemann, J. A., Timokhov, L., Gutjahr, O., and Heinemann, G.: Circulation in the northwest Laptev Sea in the eastern Arctic Ocean: Crossroads between Siberian River water, Atlantic water and polynya-formed dense water. J. Geophys. Res. Oceans, 122, 6630–6647, https://doi.org/10.1002/2017JC013159, 2017.